

# Charactersing spatio-temporal variability in seasonal snow cover at a regional scale from MODIS data: The Clutha Catchment, New Zealand

Todd A. N. Redpath[1,2], Pascal Sirguey[1], and Nicolas J. Cullen[2]

[1]National School of Surveying, University of Otago, PO Box 56, Dunedin, New Zealand
[2]Department of Geography, University of Otago, PO Box 56, Dunedin, New Zealand

**Correspondence:** Todd Redpath (todd.redpath@otago.ac.nz)

**Abstract.** A 16-year series of daily snow covered area (SCA) for 2000 – 2016 is derived from MODIS imagery to produce a regional scale snow cover climatology for New Zealand's largest catchment, the Clutha Catchment. Filling a geographic gap in observations of seasonal snow, this record provides a basis for understanding spatio-temporal variability in seasonal snow cover, and combined with climatic data, provides insight into controls on variability. Metrics including daily SCA, mean
snow cover duration (SCD), annual SCD anomaly and daily snowline elevation (SLE) were derived and assessed for temporal trends. Raster principal components analysis (rPCA) was applied to maps of annual SCD anomaly to characterise modes of spatial variability whilst preserving temporal signals. Semi-distributed analysis between SCD and temperature and precipitation anomalies allowed sensitivity of SCD to climatic forcings to be assessed spatially. The influence of anomalous winter air flow, as characterised by HYSPLIT back-trajectories, on SCD variability was also assessed. On average, SCA peaks in late June, at
around 30% of the catchment area, with 10% of the catchment area sustaining snow cover for > 120 days per year. A reduction in SCA through mid-winter, prior to a second peak in August and persistent throughout the time series is attributed to the prevalence of winter blocking highs in the New Zealand region. In contrast to other regions globally, no significant decrease in SCD was observed. rPCA identified six distinct modes of spatial variability, characterising 77% of the observed variability in SCD. rPCA and semi-distributed analysis of SCD anomalies reveal strong spatio-temporal variability beyond that associated
with topographic controls, which can result in snow cover conditions being out of phase across the catchment. Furthermore, it is demonstrated that the sensitivity of SCD to temperature and precipitation variability varies significantly across the catchment. While two large scale climate modes, the SOI and SAM, fail to explain observed variability, specific spatial modes of SCD are favoured by anomalous airflow from the NE, E and SE. These findings illustrate the complexity of atmospheric controls on SCD within the catchment and support the need to incorporate atmospheric processes that govern variability of the energy
balance, as well as the re-distribution of snow by wind in order to improve the modelling of future changes in seasonal snow.

## 1 Introduction

Globally, seasonal snow packs represent an important, yet vulnerable water resource (Barnett et al., 2005; Mankin and Diffen-baugh, 2015), and play a major role in earth-atmosphere energy exchange (Estilow et al., 2015). A decline in seasonal snow



covered area, and shifts in the timing of snow melt runoff have been observed in many regions of the world (McCabe and Clark, 2005; Lundquist et al., 2009; Estilow et al., 2015; Klein et al., 2016; Borman et al., 2018). While most observations of reduced seasonal snow covered area come from the Northern Hemisphere (Borman et al., 2018), negative trends have been observed over short time scales for both the alpine region of Australia (Bormann et al., 2012) and much of the South American

Andes (Saavedra et al., 2018). Despite the relatively recent establishment of the Snow and Ice Network (SIN) of alpine automatic weather stations (Hendrikx and Harper, 2013), long-term records of snow occurrence and persistence are rare in New Zealand (Fitzharris et al., 1999). This scarcity of data has limited the empirical characterisation of seasonal snow processes. Subsequently, little is known about the persistence of seasonal snow across large areas of the Southern Alps/Kā-Tiritiri-o-te-Moana, the variability associated with snow persistence from year to year, and the climatic influences on such variability.

Previous work to assess historical trends in snow water equivalent (SWE) in the main hydro-electric catchments of the South Island of New Zealand, which was largely model driven, revealed substantial temporal variability, but no long-term trend from the 1930's-1990's (Fitzharris and Garr, 1995). Conversely, modelling work undertaken to assess future impacts of climate change on seasonal snow in New Zealand predicts a substantial reduction overall in seasonal snow cover duration (SCD), snow proportion of precipitation and peak snow accumulation through the 21st Century (Hendrikx et al., 2012; Jobst et al., 2018).

The apparent absence of any long term trend in seasonal snow in New Zealand (Fitzharris and Garr, 1995), and the associated uncertainty resulting from a scarcity of observational data, provides clear motivation for the construction of an updated observational record of seasonal snow in New Zealand's Southern Alps. The need to understand likely future scenarios of seasonal snow cover underscores the importance of such data. Remote sensing offers a means to mitigate the limitations imposed by a scarcity of in situ observations of seasonal snow, albeit over a relatively limited time span.

Remote sensing has provided substantial advances in quantification of seasonal snow variability, with imaging sensors supporting spatial and temporal resolutions that allow a range of scales to be explored. Space-borne satellite imagers, particularly optical sensors, provide a synoptic view and have provided a step-change in capability for mapping and characterising snow covered areas, although trade-offs exist between competing resolutions (Andersen, 1982; Dozier, 1989; Nolin and Dozier, 1993; Hall et al., 2002, 2015; Rittger et al., 2013; Sirguey et al., 2009). For example, the MODerate resolution Imaging Spec-

troradiometer (MODIS), operating on board TERRA since 2000, and on AQUA since 2002, permits near-daily mapping of snow covered area (SCA) at continental to global spatial scales, despite relatively coarse spatial resolution. The advance of geostationary meteorological satellites such as Himawari 8 & 9 sees comparable spatial resolution to MODIS acquired in near real-time (Bessho et al., 2016). Multi-spectral sensors such as Sentinel-2A & B, and Landsat 8 continue to improve the temporal resolution of imagery suitable for mapping snow at resolutions of 10 – 30 m (Malenovský et al., 2012; Roy et al., 2014).

Passive and active microwave sensors offer the capacity to retrieve estimates of snow water equivalent directly from space-borne platforms, but also suffer substantial limitations, including coarse spatial resolution in the case of passive microwave sensors, and/or complications associated with complex terrain (Lemmetyinen et al., 2018). Despite the progress in mapping SCA, reliable determination of snow depth, particularly in complex terrain, remains challenging. Modern, very high resolution stereo-capable imagers show promise for retrieving snow depth over large areas, from space, although the influence of

topography on uncertainties, and complications introduced by shadows in alpine terrain demand attention (Marti et al., 2016).



Although direct observation SWE from space remains complicated, methodologies for extracting SCA from MODIS imagery can now provide relatively long records of spatio-temporal variability in SCA, itself a useful climatic indicator (Clark et al., 2009; Estilow et al., 2015). This is evidenced by the recent emergence of studies examining spatial and temporal variability in seasonal snow cover, and snow cover climatologies, across medium to large spatial scales (e.g., Gascoin et al., 2015; Saavedra et al., 2017, 2018).

This paper aims to produce a snow cover climatology for New Zealand's largest catchment, the Clutha Catchment, for the years 2000 – 2016, characterise the associated spatio-temporal variability, and refine understanding of controlling climatic processes. This is achieved by leveraging MODIS imagery to produce a 16-year time series of daily snow covered area. While the mapping of snow covered area from MODIS imagery is a mature approach (Hall et al., 2002; Sirguey et al., 2009; Dozier et al., 2009; Nolin, 2010; Rittger et al., 2013), characterising the spatial and temporal variability observed in the MODIS time series requires the application of robust geospatial and statistical techniques, specifically in the application of spatialised principal components analysis (PCA) to efficiently characterise spatio-temporal variability. Climatic controls are considered in terms of sensitivity of snow cover duration (SCD) to temperature and precipitation anomalies, standard climate indices, and variability in airflow as characterised by atmospheric particle trajectories.

## 2  The Clutha Catchment

Drained by the Clutha River (mean annual discharge ~530 m$^3$ s$^{-1}$(Murray, 1975)), and with an area of approximately 20,800 km$^2$, New Zealand's largest catchment, the Clutha is a watershed of primary importance to the both the Otago Region and New Zealand as a whole. Addressing the rapidly growing water needs within the Clutha Catchment presents a future challenge as demands for urban water supply and irrigation for agriculture and horticulture increase, alongside longstanding hydro-electricity commitments. Originating on the eastern side of the Southern Alps, much of the headwater flows are derived from rainfall and the melt of alpine snow cover, along with minor contributions from glaciers. The contribution of snow melt to the annual stream flow of the Clutha River is estimated to be 10% by the time it reaches the Pacific Ocean (Kerr, 2013), however this proportion is substantially higher for alpine sub-catchments and large inland basins, where it may exceed 30 – 50%. Importantly, the spatio-temporal variability in seasonal snow within the catchment is currently not well understood. Some large tributaries arise in the semi-arid inland basins of Central Otago, where annual precipitation may be as low as 400 mm yr$^{-1}$, an order of magnitude less than on the Main Divide (Macara, 2015). The Clutha Catchment includes the Dart, Rees, Shotover, Kawarau, Matukituki, Wilkin, Hunter, Nevis and Manuherikia rivers. Further references to specific catchments within this paper will be made to the Upper Clutha (Clutha up-stream of Cromwell, including lakes Wanaka and Hawea), Kawarau (west of Cromwell, including Lake Wakatipu), Lower Clutha (downstream of Cromwell) and the Manuherikia Basin, north-east of Alexandra (Figure 1).

The Clutha River hosts two nationally significant hydro-electric power stations at the Clyde (432 MW) and Roxburgh (320 MW) dams, and smaller hydro-electric power stations operate on tributaries. Irrigation is particularly important in the central and eastern basins of the catchment, where climate is characterised by much lower rainfall than the mountainous





regions near the main divide of the Southern Alps, or the coastal lower reaches of the Clutha (Murray, 1975; Hinchey et al., 1981; Macara, 2015). An important aspect of irrigation schemes in drought prone basins such as the Manuherikia, is the capture and storage of runoff generated by spring snowmelt for use later in summer (Hinchey et al., 1981). Two major local irrigation dams, the Falls Dam in the upper Manuherikia, and the Fraser Dam west of Alexandra, have estimated snowmelt contributions

of 15 and 20% respectively (Kerr, 2013). While water management is not a new concept in these areas, the role of snow in the hydrological cycle, and the impact of variability in the extent and duration of winter snow cover in the Central Otago mountain ranges, has received little attention. While modelling studies indicate that significant future changes in the snow hydrology of the Clutha Catchment are likely (e.g., Poyck et al., 2011; Jobst, 2017; Jobst et al., 2018), observational studies of snow hydrology have been discontinuous and mostly limited to small sub-catchments (e.g., Fitzharris et al., 1980; Fitzharris and

Grimmond, 1982; Sims and Orwin, 2011).

Seasonal snow underpins winter tourism, further contributing to the local economy (Hopkins, 2014). There are four commercial downhill ski areas, as well as a cross-country ski area, clustered in the Queenstown/Wanaka area. All ski areas have undergone exapansion works in recent years, with further expansion planned. A substantial heliski industry is also based out of Queenstown and Wanaka. The commercial ski season typically runs from mid-June to early October, however opening dates

and season length are variable, with lifts not operating until mid-July in the "winter of discontent" of 2011 (Hopkins, 2015). Whilst modelling work suggests that the duration of natural snow pack for ski areas in Otago will be substantially reduced, and snowmaking will become increasingly important with future climate change (Hendrikx and Hreinsson, 2012), industry participants perceive inter-annual variability as having a greater impact on ski seasons (Hopkins, 2014, 2015).

Within the catchment, forest cover at elevations above 1000 m is limited. According to version 4.1 of the New Zealand Land

Cover Database (Landcare Research, 2015), forest constitutes 2.84% of the total area of 6602 km$^2$ above 1000 m (Table 1) Given the difficulties associated with recovering reliable SCA estimates from MODIS in forested areas (Raleigh et al., 2013), the low proportion of forest landcover is advantageous for the application of remote sensing techniques compared to many other mid-latitude locations.

## 3 Data and method

### 25 3.1 MODIS multispectral imagery and MODImLab

MODIS is particularly useful for mapping snow covered area (Barnes et al., 1998; Hall et al., 2002; Sirguey et al., 2009; Painter et al., 2009; Rittger et al., 2013). In terms of spectral resolution, the SWIR bands 6 (1640 nm) and 7 (2130 nm) permit the implementation of of the normalised difference snow index (NDSI), and also facilitate enhanced discrimination of snow from other targets when spectral unmixing is utilised (Sirguey et al., 2009). Additionally, global daily to near-daily re-visit

times, allow temporally dynamic phenomena to be captured, while also maximising the likelihood of cloud-free retrievals. Furthermore, the spatial resolution is useful for monitoring change at regional scales whereby image fusion techniques can be employed to map snow using the 250 m resolution of MODIS bands 1 and 2, offering an improvement over the native 500 m resolution of MODIS bands 3-7 (e.g., Sirguey et al., 2008).



Gascoin et al. (2015) highlighted the suitability of MODIS derived snow cover products for characterising temporal variability in SCA, and providing insight into regional snow climatology for the Pyrenees Mountains. Among several robust methodologies for mapping SCA from MODIS imagery (Masson et al., 2018), MODImLab (Sirguey et al., 2009) is used here due to its desirable capabilities and specific performance in the Southern Alps. Snowcover mapping using MODImLab for the

Waitaki Catchment in New Zealand (immediately north of the Clutha Catchment) achieved a mean absolute error (MAE) <10% when compared with higher resolution ASTER reference products (Sirguey et al., 2009). Masson et al. (2018), while noting the difficulty of balancing high recall and high precision for snow mapping from MODIS imagery, found that overall MODImLab compared well with several alternatives when applied in the European Alps, Pyrenees and Moroccan Atlas Mountains.

MODImLab employs a spectral unmixing approach (in contrast to NDSI correlation applied elsewhere) to MODIS images

which have undergone preprocessing, including image fusion to faciliate mapping of snow at 250 m resolution, and a rigorous atmospheric and topopgraphic correction, ATOPCOR (Sirguey et al., 2008, 2009; Sirguey, 2009; Sirguey et al., 2016; Masson et al., 2018). This approach improves the performance of snow mapping in rugged alpine terrain, typical of New Zealand's Southern Alps. MODImLab operates on the basis of eight possible end-members (three being snow and one of ice), originally characterised in the New Zealand context. The pasture, rock and debris end-members are expected to align well with the

landcover classes detectable within the Clutha Catchment at the MODIS scale (i.e., > 250 m) (Table 1). Re-projection of tiles from the MODIS L1B swath is handled by the swath-to-grid module within MODImLab. Imagery was resampled to a 250 m grid, in New Zealand Transverse Mercator (NZTM) projection.

## 3.2  Snow covered area time series

The central process underlying this study was the construction of a continuous time series of daily, virtually cloud-free, frac-

tional snow covered area (fSCA) maps for the Clutha Catchment from MODIS imagery. Collection 6 (C6) MODIS/TERRA L1B swath granules were downloaded for all available observations from April 1 2000 to March 31 2016. Re-projection, preprocessing and mapping of fSCA was carried out using MODImLab (see full description in Sirguey et al., 2009), before implementing a cloud-filling algorithm to build the final 16-year daily cloud-free fSCA data cube used for subsequent derivation of metrics and analysis (Figure 2).

### 25  3.2.1  Cloud detection and gap filling

The presence of cloud, and mis-classification of cloud, or cloud shadows, as snow pixels is a common problem in multispectral remote sensing (Dozier et al., 2008). The development of a robust snow cover climatology benefits from having a cloud-free daily timeseries. This was achieved through the implementation of the time trajectory interpolation proposed by Dozier et al. (2008). The MOD35 cloud cover product (Frey et al., 2008) provided the basis for determining the cloud obscured pixels for

which fSCA had to be interpolated. Initially, those pixels flagged as "certain cloud" were masked. Sparse commission errors, often occurring at the edge of snow cover due to confusion between the spectral signature of cloud and mixed ground and snow pixels were mitigated by an erosion of the binary mask. Two dilatations were then applied to restore the main cloud structures





and conservatively extend the edges of cloudy regions. Each daily map of fSCA was combined with the associated daily cloud mask, with cloud affected pixels subesequently flagged for filling by fSCA interpolation.

Gaps resulting from cloudy pixels are then filled as proposed by (Dozier et al., 2008), by applying a smoothing spline interpolant to each pixel-time trajectory. Implementation of the cloud-filling algorithm provided temporally continuous maps of

daily fSCA. For convenience, the time series was processed into 365 day blocks, corresponding to the New Zealand hydrological year (hy, 01 April - 31 March). These annual gap-filled time series provide the basis for further analysis of snow cover climatology and variabilty.

### 3.3   SCD and derived metrics

The snow cover climatology can be considered intially in terms of the SCD for each hydrological year. That is, the number of

days on which snow lies on the ground. SCD can be considerd hypsometrically per discrete elevation bands (e.g., Gascoin et al., 2015), or spatially per-pixel, as it is presented here. Spatial determination allows SCD to be mapped across the catchment. SCD was calculated as the number of days for which a pixel exceeded a minimum fSCA to avoid skewing the time series by spurious short lived snow fall events and heavy frost detected as snow cover. In this case, the minimum threshold fSCA ($fSCA_t$) was set to 50%, consistent with the approach of Sirguey et al. (2009).

Calculating the SCD for each hydrological year from 2001-2016 (which represents the winter of the previous calendar year) also allowed the calculation and mapping of the mean ($\mu_{SCD}$) and standard deviation ($\sigma_{SCD}$) of SCD across the catchment, over the 16-year period. In order to characterise and map associated variability, the coefficient of variation ($CV_{SCD}$) was also calculated as:

$$CV_{SCD} = \frac{\sigma_{SCD}}{\mu_{SCD}}. \tag{1}$$

#### 3.3.1   Basin snow covered area

The temporal variability of snow covered area across the basin (basin snow covered area, bSCA) was assessed using several metrics (Table 2). These bSCA metrics were calculated as the number of days for which snow covered pixels ($fSCA \geq fSCA_t$) comprised at least 10, 15 and 20% of the total catchment area. Area thresholds of 10, 15 and 20% were chosen based on basin hypsometry, and informed by the mean snow covered area. For each hydrological year, the total number of days

meeting each bSCA threshold was calculated. Linear regression was applied to the annualised time series for each metric and each threshold to test the presence of any significant temporal trends.

#### 3.3.2   SCD anomaly

For each hydrological year, the anomaly in SCD was calculated on a pixel-wise basis as:

$$anom_{SCD} = SCD_n - SCD_\mu, \tag{2}$$

where $SCD_n$ is the SCD map for a given year, and $SCD_\mu$ is the mean SCD of all years. The magnitude of $anom_{SCD}$ was mapped in absolute terms. A total of 16 maps of $anom_{SCD}$ were derived, for the hydrological years 2001-2016 inclusive.



### 3.3.3 Snowline elevation

The snow line elevation (SLE), which may be defined as the minimum elevation at which relatively contiguous snow cover is present within a landscape is a useful metric for characterising the state of seasonal snow for a given epoch (Krajčí et al., 2014; Drolon et al., 2016). In simple terms, the occurence of snowfall and the development of a snowpack is expected to be governed

by temperature and precipitation, and this often generally results in a well defined snowline in New Zealand (Barringer, 1989). The method developed by Krajčí et al. (2014) was implemented to determine the SLE from fSCA maps and a corresponding digital elevation model (DEM). Where cloudfilling approaches are implemented with MODIS snow cover time series, this approach provides a continuous times eries of daily SLE. Deriving snow line elevation is an iterative process, where for a test elevation ($z_i$) the number of snow-covered pixels ($P_S$) below, and the number of snow-free pixels ($P_L$) above is minimised.

This is determined by calculating an $F$ value for each test elevation as follows (Krajčí et al., 2014):

$$F(z) = P_{S(Z)} + P_{L(Z)}, \tag{3}$$

$$SLE = \underset{z}{\arg\min} \, F(z). \tag{4}$$

In the case of the Clutha Catchment, $F(z)$ is calculated from 300 to 2700 m at 10 m increments. Elevation data were sourced from the NZSoSDEM, a freely available 15 m resolution DEM (Columbus et al., 2011), resampled to 250 m to match with

the fSCA maps. SLE was determined using the 16-year gap-filled time series derived from MODImLab fSCA products. Pixels were considered snowcovered for the purpose of deriving SLE when pixel $fSCA \geq fSCA_t$.

## 3.4 Characterising spatio-temporal variability

### 3.4.1 Spatialised principal components analysis

Principal component analysis (PCA) reduces the dimensionality of multivariate datasets by determining linear combinations

of variables that provide a number of de-correlated principal components (PCs), each characterising a proportion of the total variance in the underlying data (Dunteman, 1989; Jackson, 1991). This approach has a long history of use in climatological research for characterising spatial patterns in the distribution of key parameters such as temperature, precipitation and atmospheric pressure (Tait et al., 1997), sea ice concentration (Baba and Renwick, 2017) and ice shelf behaviour (Campbell et al., 2017). In these applications, PCA typically reveals the spatial structure of important, and often persistent, features within

spatially distributed observations. The temporal component of variability, however, may not be readily interpreted from this "atmospheric science PCA" (Demšar et al., 2012).

When applied to spatio-temporal data, such as time series of raster maps capturing a spatially continuous variable, each principal component is a map representing a spatial mode of variability occurring within the time series. The PCs are ordered according to the proportion of overall variance they explain. Conceptually, this approach, which can be considered as "raster

PCA" (rPCA) (Demšar et al., 2012) is analogous to the application of PCA to multispectral imagery, where PCs are determined in a temporal, rather than spectral, attribute space. Spatialisation is a product of the georeferenced raster datasets, and PCs are calculated only in the attribute space, so geographic effects are not accounted for (Demšar et al., 2012). A key benefit of the





rPCA approach is that, when applied to a raster time series, it allows the occurrence of specific modes of spatial variabilty to be considered as a function of time. Thus, the spatio-temporal variability within the data is preserved and can be further scrutinised and interpreted.

Here, rPCA was applied to the time series of SCD anomaly maps. The resulting PCs provide an indication of the spatial and temporal variability in SCD anomaly. PCA was carried out using Matlab®. The single value decomposition algorithm was applied to the the time series of 16 SCD anomaly maps, yielding as many PCs.

### 3.4.2  Temporal dynamics of spatial modes

Each PC from the rPCA described in Section 3.4.1 reveals spatial modes of variability. In turn, the respective loadings of each PC can be used to identify years which share similarities in the spatial expression of their SCD anomalies. Comparing years based on their PCA signatures was done by considering the loadings of the first six PCs, which explained 77% of the total variance in the spatio-temporal dataset. The PCA provided a six-dimensional loading space, within which the similarity between years was measured by the Euclidean distance and k-means clustering.

### 3.4.3  Mountain range analysis

An alternative, semi-distributed approach was condsidered for characterising spatial variability in SCD. Addressing space via individual mountain ranges, provided a basis to explore spatial gradients in SCD variability across the catchment. This approach allows the catchment to be treated in a way that may be more readily understood by stakeholders, potentially including industrial and agricultural water managers and consumers, ski area operators and the general public. Such stakeholders typically consider the snow resource at a sub-catchment mountain range scale. The goal of this analysis was to determine whether mountain ranges provide a useful geogrpahic unit for capturing and characterising spatial variability. The catchment was subdivided into 36 different mountain ranges, based on the naming conventions provided by the 1:50,000 scale NZ Geographic Names topographic dataset (LINZ, 2018). All land areas below the median winter snow line elevation (as calculated in Section 3.3.3) of 1270 m were excluded. Then, areas of mountain ranges above this elevation were manually digitised using a DEM (Columbus et al., 2011) and stream centreline vector data (Snelder et al., 2010) as guidance (Figure 3). Since the Main Divide of the Southern Alps is considered to provide a baseline for climatic gradients in the east of the South Island, owing to predominant westerly air flow and the substantial orographic effects of the Southern Alps (Sinclair et al., 1997; Jobst et al., 2018), the distance from each range to the main divide of the Southern Alps was calculated using the *Near* tool in ArcGIS v10.3. Mean annual SCD anomalies for each range were then extracted from the maps calculated in Section 3.3.2. Ranking each range in terms of proximity to the main divide then provided a basis for examining spatial trends in the SCD anomaly across the catchment from year to year.



### 3.5 Climatic influences on snow cover variability

Beyond characterising the snow climatology for the Clutha Catchment, relating spatio-temporal variability in SCD to climatic variability is central to better understanding climatic influences on seasonal snow in New Zealand. This was done firstly by assessing the sensitivity of the annual SCD anomalies to annual temperature and precipitation anomalies. Furthermore, the

role of atmospheric circulation was assessed by considering airflow as characterised by multiple atmospheric back trajectories generated for the study period using the Hybrid Single-Particle Lagrangian Integrated Trajectory (HYSPLIT) model (Stein et al., 2015). Airflow analysis was motivated in part by the suggestion that increased frequency of southerly airflow can significantly suppress temperatures in New Zealand (Dean and Stott, 2009), but also because variability in airflow can be expected to produce spatial variability in the delivery of precipitation to the Clutha Catchment.

Several climatic datasets were acquired to investigate the role and relative influence of climatological processes in controlling the variability of seasonal snow, namely:

1. Gridded temperature and precipitation fields, comprising daily $T_{max}$, $T_{min}$, $T_{mean}$ and $P_{tot}$ for the period 2000-2016, interpolated at a resolution of 250 m, coincident with the pixel size of comparable MODIS datasets, produced as detailed in Jobst et al. (2017) and Jobst (2017).

2. Global gridded (2.5°) NCEP/NCAR meteorological reanalysis data (Kalnay et al., 1996), used as input fields for the HYSPLIT (version 4) model. Data were acquired from the Climate Data Center (CDC) via the ftp module within the HYSPLIT software.

#### 3.5.1 Spatial anomalies for temperature and precipitation

The daily fields of mean temperature and precipitation described above were used to calculate pixel-wise mean values for

each hydrological year, as well as for each winter period. Winter was defined as being from May 01 – October 30, the period during which the majority of substantial snowfall events occur. Pixel-wise anomalies in annual mean temperature and total precipitaion were calculated to produce a series of 16 anomaly maps for each variable, which were compared directly with the SCD anomaly maps. No significant correlation existed between temperature and precipitation anomalies.

The sensitivity of SCD anomalies to temperature and preciptation anomalies was assessed in a semi-spatially distributed way.

Regression analysis was carried out between the annual SCD anomaly and temperature and precipitation anomalies averaged over each mountain range. Sensitivity of SCD to temperature and precipitation variability could then be quantified and its significance assessed for each mountain range across the catchment.

#### 3.5.2 Synoptic scale airflow variability

HYSPLIT is a Lagrangian air parcel trajectory and dispersion model (Stein et al., 2015). While it offers complex simulations

of particle atmospheric transport, dispersion, chemical transformation and deposition, the trajectories modelled by HYSPLIT





provide a means to characterise airflow, and associated airmass characteristics. Here, HYSPLIT was used to determine and analyse the origin and trajectory of air parcels arriving to the Clutha Catchment throughout the study period.

The geographic end-point for the trajectories was a location near the summit of the Pisa Range (Figure 3), with elevation set to 2000 m. This location was close to the geographic center of the Clutha Catchment (-44.90º S, 169.16º E, Figure 3) and

therefore expected to provide a suitable characterisation of modes of air flow affecting the catchment throughout the year. Back trajectories were calculated over 120 hours for each day of the study period, with four temporal end-points per day (0000, 0600, 1200, 1800 hrs). In total this provided 1460 trajectories per year.

HYSPLIT trajectories were used to construct indices of airmass origin. For four timesteps, namely -96, -72, -48 and -24 hours, the trajectory direction relative to the end-point was determined and binned according to eight directional classes (i.e.,

N, NE, E, SE, S, SW, W, NW). This allowed the mean frequencies of air parcel origin direction, and relative frequencies for each winter (01 June – 30 September) period to be calculated. The relative frequency distribution for each winter then provided a means to characterise air-flow characteristics for each winter.

Relationships between the annual loadings of each principal component of SCD and the relative frequency of air parcel origin direction were assessed by multiple linear regression. Because of the large number of variables (8) relative to the number of

years for which data exists (16), only four directions exhibiting the greatest inter-annual variability were included in multiple linear regression to mitigate against under-determination. It comes that, for each principal component ($n$), and each year ($y$), the linear model took the form:

$$\lambda_{n(y)} = \beta_{0(y)} + \beta_{1(y)}x_{1(y)} + \beta_{2(y)}x_{2(y)} + \beta_{3(y)}x_{3(y)} + \beta_{4(y)}x_{4(y)}, \tag{5}$$

where $\lambda_{n(y)}$ is the loading of the $n^{th}$ PC for that year, and $\beta_{0(y)}, ..., \beta_{4(y)}$ and $x_1, .., x_4$ are the SCD sensitivity to, and relative

frequency of air parcels originating to the NE, E, SE and S. This analysis allowed the contribution, importance, and significance of anomalous airflow directions to the modes of spatial variability in SCD to be assessed.

### 3.5.3   Assessing the role of climate modes

Commonly, short- to medium-term variability in climate and cryospheric processes in mid-latitude regions has been considered in terms of large scale climate modes such as the Southern Oscillation Index (SOI), the Southern Annular Mode (SAM)

and the Pacific Decadal Oscillation (PDO) (e.g., McKerchar et al., 1998; Fitzharris et al., 2007; Kidston et al., 2009; Purdie et al., 2011; Sirguey et al., 2016). Periods of negative SOI phase are expected to correspond with cooler periods of increased solid precipitation in New Zealand, and have been associated with positive mass balance for glaciers in the Southern Alps (McKerchar et al., 1998; Lamont et al., 1999; Fitzharris et al., 2007; Purdie et al., 2011), yet its inclusion as a predictor variable does not necessarily increase the skill of hydrological forecast models (Purdie and Bardsley, 2010). Here, mean winter values

of the SOI and SAM indices were compared to each of the temporal metrics characterising snow cover within the catchment (Table 2) via correlation analysis and linear regression. The PCA analysis proposed by this study provides new insights into the spatial expression of snow cover variability that also deserves further scrutiny. Subsequently, the influence of large scale



circulation indices on spatial modes of snow cover was explored by analysing the correlation between the SOI and SAM and annual PC loadings.

## 4  Results

### 4.1  Snow cover climatology for the Clutha Catchment

The typical snow cover climatology for the Clutha Catchment is demonstrated by $\mu_{SCD}$ over the period 2000-2016. The first-order control of elevation on snow persistence is obvious in Figure 4(a). On average, an area of 2218 km$^2$, or 10% of the total catchment area, maintains snow cover for 120 days a year or more. In general, snow cover is most persistent on terrain above 1500 m a.s.l., and on slopes with Southern aspect.

The mean SLE during the winter period was found to be 1263 m a.s.l, this corresponds with an area of 3180 km$^2$ and is
substantially higher than the mean catchment elevation of 615 m, and . On average, the snowline is at or below this elevation from mid-June until mid-August (Figure 5). A small area of 76 km$^2$ maintains permanent snow cover (SCD > 360 days). This corresponds with a maximum SLE of around 2200 m a.s.l. Terrain that maintains snow-cover year round is largely restricted to glacierised regions within a few kilometres of the main divide, and appears to be strongly controlled by aspect.

The onset of the snow season is typically rapid, with SCA increasing, and SLE lowering abruptly in the early winter, reaching
their maximum and minimum, respectively, in mid-June (Figure 5). At the catchment scale, both SCA and SLE feature a mid-winter depletion/rising respectively, before SCA increases and SLE decreases again in late winter/early spring. Maximum SCA typically reaches > 35% of the catchment area during June, yet the depletion curve is punctuated by local maxima. On average, the first of these occurs in early July, and the second in mid-August. These features are persistent throughout the 16 – year study period. In addition, a minima in variability of daily SCA occurs in late April, following an increase in variability due
to early snowfall over the preceding month. Variability in basin SCA and SLE is greatest during the May-July period, and converges towards the mean through spring and early summer.

### 4.2  Topographic controls on snow duration and variability

Figure 6 shows the overall control of aspect and elevation on SCD and its variability. Most pixels exhibiting $\mu_{SCD}$ >120 days have a southerly aspect between 90 and 315º, and elevation in excess of 1500 m. The role of aspect strengthens as elevation
decreases. The variability in SCD is more pronounced on northerly aspects between 315 and 90º, across all elevations. In contrast, the $CV_{SCD}$ rarely exceeds 100% for pixels with S to SW aspect between 135 and 270 degrees, but may exceed 200% for pixels with northerly aspect between 315 and 90º (Figure 6).

$\mu_{SCD}$ is positively, though not linearly, correlated with elevation. The rate of increase of $\mu_{SCD}$ with elevation is higher on southerly than northerly aspects (Figure 6). A $CV_{SCD}$ <10% is generally restricted to southerly aspects above 1500 m.
Elsewhere, variability in SCD is relatively large. $CV_{SCD}$ increases markedly at lower elevations reflecting the relatively short $\mu_{SCD}$ and associated large inter-annual variability.





### 4.3 Spatio-temporal variability in seasonal snow cover

The map of $CV_{SCD}$ shown in Figure 4(b) highlights the magnitude of spatio-temporal variability associated with SCD in the Clutha Catchment. A first order control of elevation on $CV_{SCD}$ is apparent, but further spatial variation exists beyond the basin hypsometry. Areas with $CV_{SCD} \leq 5\%$ were only present on and near the main divide, while $CV_{SCD}$ is much greater

elsewhere in the catchment At moderate to high elevations, $CV_{SCD}$ in the western part of the catchment was generally less than 50%. In the eastern part of the catchment, $CV_{SCD}$ was generally greater, but this reflects the basin hypsometry to an extent. Ultimately, the large $CV_{SCD}$ observed across most of the catchment reflects the temporally dynamic nature of snow cover across the Clutha Catchment.

### 4.3.1 Temporal variability in catchment-wide snow cover and snowline elevation

The full time series of daily SCA and SLE, and associated departures from their series means reveals the extent of temporal variability in seasonal snow at the catchment scale (Figure 7). Substantial departures from the mean throughout the time series highlight the strong inter-annual, as well as shorter term variability in the occurrence, persistence and disappearance of seasonal snow within the catchment. Large, yet short-lived, perturbations occur due to unseasonable snow events, while sustained positive and negative anomalies occur at seasonal scales during winter. The maximum observed SCA was 65%,

and occurred on June 18 2002. The only other occasion on which SCA exceeded 60% of catchment area was June 21 2013. Notably, the years that feature the largest maximum SCA values are not always associated with a sustained positive anomaly in catchment-wide SCA.

The considerable temporal variability associated with SCA is further demonstrated by the bSCA metrics plotted in Figure 8. The total number of days each year with bSCA exceeding 10, 15 and 20% is highly variable from year to year, and did not

reveal any significant trend over the 16-year time series. The lowest returned *p*-value (0.23) was associated with a negative trend for the end date of temporally continuous 15% bSCA. Typically, 10% bSCA is achieved by June, with the latest onset date of July 06 occurring for hy 2012. Generally, 10% bSCA is sustained through September 30, resulting in a mean 120-day duration. The earliest end date for 10% bSCA, of September 08 occurred for hy 2006. Several years saw 10% bSCA extend through, or beyond, October 31. For hy 2010, an early onset of continuous 10% bSCA (07 May), combined with a loss date of

30 September, resulted in the longest continuous duration of 10% bSCA, of 176 days.

Both 15 and 20% bSCA saw much more variability, with the period of 20% bSCA being discontinuous for most winters. Onset of both 15 and 20% bSCA typically occurred within a few days of 10% bSCA, however there was usually considerable lag between the end date of 15 and 20% bSCA and that of 10% bSCA.

The winter SLE also exhibited significant intra- and inter-annual variability, both in terms of the mean (median) winter SLE

of 1263 m (1270m̄), and the range of observed SLE (Figures 7 & 9). The largest ranges of observed winter SLE (e.g.,hy 2012), were associated with years when the onset of winter snow cover was delayed. This scenario reflects an over-representation of high SLE through June, which is the month when the lowest SLE values are usually observed across the entire record. Similarly to SCA, no significant temporal trend was detected in SLE over the 16-year record.





### 4.3.2  Modes of spatial variability

Mapping the principal components of SCD anomaly revealed distinct modes of spatial variability (Figure 10). 77% of the total observed spatial variability in annual SCD anomaly was explained by the first six principal components (Figure 11). In turn, each annual SCD anomaly map can be efficiently reconstructed via linear combination of each map of PC scores. As

such, the loadings of the combination quantify the relative contribution of each mode to the SCD map. Since PC scores are not standardised, their unit matches that of the SCD anomaly (i.e., day). Where loadings are positive (negative), a positive PC score will propagate positively (negatively) into the SCD anomaly, with the inverse being true for negative PC scores. PC1 (39% of total variability), is characterised by a negative PC score across the whole catchment that reflects the first order control imposed on SCD by elevation (Figure 10). PC2 (14.8% of total variability) features an annular pattern expressed across much of the

catchment, where negative PC scores are associated with high elevation areas, while positive scores occur for lower elevation areas, and in the east of the catchment. Overall, PC2 highlights the existence of a spatial trend in variability that departs from the topographic control characterised by PC1. PC3 (8.05% of total variability) further stresses a spatial trend, with negative PC scores in the west, transitioning to positive scores in the Manuherikia (north-east) region of the catchment, and for higher elevations in the central part of the catchment (e.g., the Pisa Range). PC4 (6.43% of total variability) also exhibited spatial

structure, with positive PC scores in the north of the catchment, and negative scores dominating south of the Kawarau and Manuherikia rivers. Explaining 5.12% of total variability, PC5 shows that SCD anomalies are further modulated by spatially structured contrasts between negative PC scores in the central part of the catchment, and positive scores through the western, northern, and eastern margins. Finally, PC6 (3.8% of total variability) exhibits weaker spatial structure, with positive PC scores through most of the western part of the catchment, and at low to moderate elevations towards the east balanced by negative

scores at higher elevations in central and eastern areas. The magnitude of scores was reduced for PC6 relative to PCs 1 – 5. PCs 7 – 16 each explained less than 3% of total variability.

### 4.3.3  Dynamics of spatio-temporal variability

The attribute space of annual PC loadings facilitated the comparison and grouping of years in terms of the spatial structure of their SCD anomaly. The two most similar years, in terms of PC loadings, were 2015 and 2006, while the two most dissimilar

were 2010 and 2007 (Figure 12). PC1 (38% of spatial variability), carried relatively strong positive loadings for hydrological years 2002, 2006, 2012, 2014 and 2015, while negative loadings for PC1 were associated with 2003, 2005, 2010 and 2016. The relatively large pair-wise distances between PC loadings for most years (Figure 12) illustrates that, over the 16 – year period, the spatial structure of SCD anomaly across the catchment is rarely repeated, except in 2006 and 2015. As a result, groups identified by k-means clustering of their PC loading signature are potentially weak. Assignment into four clusters performed

well at grouping the most similar years together, whilst mitigating against single-member clusters, yet spurious members remain in all clusters (Table 3 and Figure 12).



### 4.3.4 Spatio-temporal variability in SCD across mountain ranges

A semi-distributed analysis of the SCD by mountain range provides further insight into spatial variability of SCD within the catchment. Since the mean elevation varies for each mountain range, the influence of elevation on SCD could be assessed. The mean annual SCD of individual mountain ranges is shown in Figure 13, which demonstrates that average SCD is a function of

elevation, as well as proximity to the main divide. For ranges within 60 km of the main divide, a significant positive correlation exists between mean range elevation and SCD, with the modelled relationship predicting an elevation of 2298 m for perennial snow cover. Beyond distances of 60 km from the main divide, however, the relationship weakens substantially and loses significance (Table 4). The deterioration of the relationship between elevation and SCD for ranges distant from the main divide suggests that the influence of temperature on snow persistence is substantially reduced in these cases.

For four out of the 16 winters analysed (25%), the sign of the anomaly was out of phase across the catchment. For the winters of 2000 and 2003, the SCD anomaly was positive in the west, and negative in the east, while the winters of 2008 and 2010 were negative in the west and positive in the east. These similarities are consistent with these pairs of years being grouped together on the basis of PC loadings. The spatial gradients of SCD anomaly observed for these years, from west to east, are consistent with the derived principal components. PCs two and three, in particular, characterised the east-west gradient

in SCD anomaly which typifies these "out of phase" years. Both of these principal components expressed an east-west contrast in SCD anomaly, with PC2 emphasising an inverted topographic dependency. The winter of 2001 loaded negatively on PC2, while the winter of 2003 loaded negatively on both PCs 2 and 3. Conversely, the winters of 2008 and 2010 loaded positively on both PCs 2 and 3, with 2010 being the only winter to load strongly on both of these PCs. The consistent signal between PC signatures and the range anomaly approach demonstrates that both the fully distributed and range-based semi-distributed

approaches are consistent in detecting variability in spatial contrasts. It also highlights that snow cover conditions across the Clutha Catchment is often spatially out of phase, with first order control by elevation being of diminished importance. This out-of-phase behaviour, and indeed the variability in magnitude anomaly even when the entire catchment is in phase, confirms that processes at the sub-catchment scale are important in controlling variability in seasonal snow processes, and that influence of, and sensitivity to competing climatic forcings varies across the catchment.

## 4.4 Climatic influences on SCD

### 4.4.1 Sensitivity of SCD to temperature and precipitation variability

Regression analysis between SCD, winter temperature, and winter precipitation anomalies revealed a variable response across the catchment. Sensitivity to temperature and precipitation variability is expressed as days (d) per degree Celsius, or mm, respectively. Overall, relationships between temperature/precipitation anomalies and SCD anomaly, were found to be significant

($p < 0.05$) across the catchment, yet both regressions featured high dispersion, with $R^2 = 0.19$ for temperature and $R^2 = 0.11$ for precipitation (Table 5). This dispersion was found to be due to contrasting climatic sensitivity across the catchment. For temperature, 50% of ranges exhibited a statistically significant relationship between SCD and temperature anomaly. For all ranges, $\beta$ was negative, with the greatest sensitivity of $-53.6$ d ºC$^{-1}$ occurring for the Ailsa Range. For precipitation, a signif-



icant relationship with SCD was found for 39% of ranges. Values of $\beta$, were always positive, with a maximum sensitivity of 0.11 d mm$^{-1}$ occurring for the Eyre Mountains.

Sensitivity of SCD anomaly to both temperature and precipitation anomalies exhibited significant spatial trends (Figures 15 & 16). Maximum temperature sensitivity occurred on and near the main divide, reducing at a rate of $-0.43$ d °C$^{-1}$ km$^{-1}$

eastward from the Main Divide, concurrently with decreasing R$^2$ and increasing $p$-values. The relationship between temperature anomaly and SCD anomaly becomes insignificant beyond distances of 55 km from the main divide. For precipitation, a more complex spatial trend emerged, whereby the relationship with SCD anomaly was weak and insignificant for ranges on and near the main divide, before becoming significant and strengthening with distance between 20 and 55 km, where the strongest significant relationship occurred. From this point, sensitivity to precipitation, and the strength of the relationship,

decreased. The relationship between precipitation and SCD anomaly was insignificant at distances greater than 80 km from the main divide.

### 4.4.2  Influence of synoptic scale variability

Both the SOI and SAM varied from year to year, yet notable events were limited to the strong La Niña of 2011. Relationships between basin snow cover metrics and the SOI and SAM were weak and insignificant (Table 6). The lowest observed $p$-value

(0.16), was associated with a weak positive relationship between SAM and the total number of days of 15% bSCA. Given the high inter-annual variability of bSCA relative to a lack of notable SOI or SAM events, any signal within these metrics is too weak to reveal any significant relationship with these indices over the 16-year period.

Similarly, there was no significant correlation detected between the SAM and the yearly loadings of principal components one to six. In the case of SOI, PC2 displayed a relatively weak positive relationship ($p = 0.06$, R$^2 = 0.23$) and PC5 a relatively

weak negative relationship ($p = 0.07$, R$^2 = 0.21$). These results suggest that the positive expression of PC2 is likely to be favoured during periods of positive SOI phase (La Niña), and PC5 during periods of negative SOI phase (El Niño). Furthermore, the enhanced signal provided by the PCs in this case highlights the benefit of utilising PCA to leverage the spatial structure of the SCA time series.

Comparison between the most anomalous years in terms of airflow, as characterised by HYSPLIT, and principal component

loadings, revealed associations between airflow and SCD anomaly. This is demonstrated for hy 2011, which was characterised by a substantial positive anomaly in easterly airflow. This corresponded with large positive loadings for both PC2, and PC3, the only year for which this occurred. Multiple linear regression between loadings for PCs one to six and relative frequency of air mass origins for 24-96 hour periods revealed strong correlations in only a few cases (Table 7). The associations that emerge from this analysis suggest that PC3 is associated with positive anomalies in easterly airflow, and negative anomalies

in south easterly airflow over periods of 24 hours, and positive anomalies in easterly airflow over 48 hours. PC2 was most strongly associated with negative easterly and positive south-easterly and southerly airflow anomalies over 72 hour periods, as well as positive southerly airflow anomalies over 96 hour periods. PC1 was most strongly associated with positive anomalies in north-easterly airflow over 96 hour periods.





Comparison between the most anomalous years in terms of airflow, as characterised by HYSPLIT, and principal component loadings, revealed associations between airflow and SCD anomaly. This is demonstrated for hy 2011, which was characterised by a substantial positive anomaly in easterly airflow. This corresponded with large positive loadings for both PC2, and PC3, the only year for which this occurred. Multiple linear regression between loadings for PCs one to six and relative frequency

of air mass origins for 24-96 hour periods revealed strong correlations in only a few cases (Table 7). The associations that emerge from this analysis suggest that PC3 is associated with positive anomalies in easterly airflow, and negative anomalies in south easterly airflow over periods of 24 hours, and positive anomalies in easterly airflow over 48 hours. PC2 was most strongly associated with negative easterly and positive south-easterly and southerly airflow anomalies over 72 hour periods, as well as positive southerly airflow anomalies over 96 hour periods. PC1 was most strongly associated with positive anomalies

in north-easterly airflow over 96 hour periods.

## 5   Discussion

### 5.1   Spatial trends in seasonal snow cover

First order topographic controls were found to exert substantial influence over mean SCD. Overall, SCD is strongly linked to elevation and aspect. This relationship was strongest on and near the main divide, with the relationship between elevation and

mean SCD breaking down in the fault block ranges located >60 km east of the main divide, as shown in Figure 13. Maximum SCD, and the only regions of perennial snow cover coincide with those areas of highest terrain and maximum mean annual precipitation along and near the Main Divide of the Southern Alps.

    The deterioration in the relationship between mean SCD and elevation for ranges south-east of the Main Divide suggests that the role temperature as a primary control on snow persistence for this region weakens. Although not specifically studied

here, a reduced frequency of cloud cover in the lee-ward rain shadow of the Main Divide may increase the relative importance of short-wave radiation in controlling ablation processes on eastern ranges. While the western ranges are alpine in nature, with high relief, the eastern ranges are characterised by high plateaus, with maximum elevations of 1400-1800 m. These plateaus are dissected by shallow basins and steep sided gullies, the floors of which are typically above the winter snowline. This morphology, combined with exposure to persistent winds is conducive to preferential accumulation during redistribution

events, promoting the development of deep snowpacks which may persist much longer into spring than would otherwise be expected (Redpath et al., 2018). The relative importance of competing climatic influences on SCD is discussed further in the following sections.

    Overall, variability relative to the mean snow cover climatology is found to be high. The coefficient of variation is negatively correlated to elevation, and greatest for pixels with aspect N-NE, consistent with the anticipated influence of topographic

variables as first-order controls. The fact that much of the catchment exhibits $CV_{SCD} > 100\%$ reflects the basin hypsometry. Most of the catchment remains below the mean winter snowline elevation, yet is still subject to occasional, often short-lived, snow fall events. A relatively large area of the catchment lies at elevations between 1000 and 1500 m and has $CV_{SCD} > 0.5$,





representing a large marginal snow season zone, where snow cover is often transient. Much of this area is in the eastern part of the catchment (e.g., the Hawkdun Range), where substantial inter-annual variability in SCD is observed.

## 5.2 Temporal trends in seasonal snow cover

Since 1909, annual mean temperature in New Zealand has increased at a rate of 0.91°C c$^{-1}$, although this rate is reduced

for the southern South Island (0.58°C c$^{-1}$ at Dunedin) (Mullan et al., 2010) and climate change is expected to result in a reduction in both the duration of snow cover, and peak snow depth in the South Island of New Zealand (Hendrikx et al., 2012). The 16 – year time series presented here, however, remains largely dominated by inter-annual variability, concealing any temporal trend that may exist in SCD or SLE. This indicates that for the 2000 – 2016 period, short term variability in seasonal snow exceeds any longer term signal driven by climate change, which contrasts a tendency toward warmer than average air

temperatures within the Clutha Catchment (Figure 18). While this daily observational record of snow presence remains short, it is consistent with earlier findings of Fitzharris and Garr (1995), whose snow storage index reconstructed by temperature-index models (in turn driven by sparse observations) for the major hydro-electric catchments of the South Island showed significant inter-annual variability, but no long term trend over the period 1931 – 1993. Subsequently, the Clutha Catchment exhibits contrasting behaviour to many other regions globally, where reductions in seasonal snow, and shifts in the timing of

seasonal snow processes have been observed (e.g., McCabe and Clark, 2005; Lundquist et al., 2009; Estilow et al., 2015; Klein et al., 2016; Borman et al., 2018). This includes other regions in the Southern Hemisphere, where in the Snowy Mountains of Australia a significant reduction in snow season length has been detected from MODIS data over the period 2000 – 2010 (Bormann et al., 2012). Similarly, the South American Andes have undergone a spatially variable but significant reduction in snow season length of 2 – 5 d yr$^{-1}$ between latitudes 29 and 36° S over the period 2000 – 2016 (Saavedra et al., 2017).

Despite the large observed inter-annual variability, the most extreme years in the current record can provide insight into the potential impact of future warming. The greatest reduction in snow cover across most metrics considered here occurred for hy 2006, during which SCD depletion was spatially consistent across the catchment (strong positive loading on PC1). This year represents an anomalously warm winter within the catchment, where temperatures at Queenstown Airport were 1°C warmer than the 2000 – 2016 average (NIWA, 2018), and was characterised by high temperatures and higher than normal sunshine

hours across much of the lower South Island (Salinger and Burgess, 2005). Relative to the 1986 – 2005 period, winter warming of 0.7 – 1.2°C is projected for the Otago Region for the period 2031 – 2050 (Mullan et al., 2016). Considering the impact of temperature alone, the probability of poor snow seasons such as that seen for hy 2006 is likely to increase.

## 5.3 Spatio-temporal variability in seasonal snow cover

The relatively large number of principal components required to explain the main share of observed variability, and the high

dissimilarity between years in terms of PC loadings, confirms the high degree of spatio-temporal variability within the time series. PC1 represents the largest share of variance occurring in SCD and is explained by elevation, with most variance occurring in the areas where seasonal snow mostly occurs and persists. PC2, however, reveals an annular pattern, whereby negative variance occurs at high elevations, and positive variance at low elevations. The maximum positive loading on PC2 occurred for





hy 2012, which also coincided with the latest observed onset date for 10% bSCA and an extended period of anomalously high SLE. This characteristic in SCA and SLE was eventually reversed, with the latter part of the winter featuring a sustained period of suppressed SLE and higher than average SCA. Ultimately, this resulted in a longer than usual SCD at low elevations, despite the reduced SCD at high elevations due to an absence of autumn and early winter snow. The two most similar hydrological

years in terms of PC loadings were 2006 and 2015. Both featured strong positive loadings on PC1, and negative loadings on most of the remaining PCs except for PC5. This resulted in strongly negative SCD anomalies for both years making them the two lowest ranked observed snow seasons across the catchment across the majority of metrics. Strong negative SCD anomalies affected all mountain ranges for these years.

Four years exhibited strong spatial gradients across the catchment, with an east-west gradient in positive to negative SCD

anomalies for 2001 and 2003, and a west-east gradient for 2009 and 2011. While each pair shared similar loadings for some PCs, their overall pair-wise signature difference was still relatively high. This observation highlights that even for years that appear to be similar in terms of spatial variability in SCD anomaly, the propagation of spatial variability across the catchment remains complex and highly variable from year to year. Notably, the occurrence of an east-west gradient in SCD anomaly, and positive loadings on PCs 2, 3, 4 and 5 indicate that the typical climatic gradient (west-east) across the catchment can be

modulated, or even reversed, in terms of SCD. The spatial structure evident across the catchment within PCs 2-6 also indicates that in terms of seasonal snow the Clutha Catchment departs substantially from behaving as a single climatic unit.

## 5.4   Climatic influences on observed snow cover variability

### 5.4.1   Influence of winter blocking highs

A persistent feature throughout the 16-year time series was the observed mid-winter decrease in bSCA, which on average

occurs during July, before SCA peaks again in August. This feature of the snow cover climatology aligns with a winter positive air pressure anomaly, which peaks in July (Figure 18). The occurrence of winter anticyclones is a common feature in the New Zealand region, and these may commonly become persistent and lead to strong blocking (Sinclair, 1996). Under such conditions, the frequency of precipitation events (i.e., snowfall), particularly those delivered by the passage of fronts, will be reduced, while at the same time, solar irradiance at the surface will be high under clear sky conditions. Together, these factors

will reduce accumulation and enhance ablation, allowing a depletion in SCA before unsettled conditions in late winter and early spring promote renewed accumulation and bSCA increases again ahead of spring melt.

### 5.4.2   Temperature and precipitation sensitivity and associated spatial variability

Temperature was found to have a more significant influence on SCD anomaly than precipitation, although sensitivity to both factors was spatially variable. The sensitivity of SCD anomaly to temperature was greatest on and near to the main divide, where

sensitivities as high as $-53$ d °C$^{-1}$ were observed, accompanied by significant R$^2$ values of $0.39 - 0.74$. Sensitivity decreased to the east, with the weakest relationship of $-6$ d °C$^{-1}$ (R$^2 = 0.01$, $p = 0.78$) observed for the Dunstan Mountains. Enhanced sensitivity to temperature on and near the Main Divide reflects the influence of orographic processes, with the rain/snow





temperature threshold and associated albedo feedback playing important roles in controlling accumulation and ablation in this cloudy, high precipitation environment. Conversely, increased incident solar irradiance through the central part of the catchment (Macara, 2015), and an increased proportion of precipitation delivered via cold fronts may act to reduce the importance of air temperature in controlling snow accumulation and ablation. For the Brewster Glacier, located west of the Main Divide and just

outside the Clutha Catchment, Cullen and Conway (2015) reported that net radiation dominated melt at annual timescales, but the role of cloud cover in enhancing sensitivity to air temperature at Brewster Glacier has also been recognised (Conway and Cullen, 2016). The Pisa Range, near the geographic center of the catchment, has a temperature sensitivity of -13.28 d ℃$^{-1}$ ($R^2 = 0.06$, $p = 0.38$), which is broadly consistent with the findings of Sims and Orwin (2011) where net radiation provided 40% of melt energy, compared to 34% from sensible heat fluxes. Conversely, further north in Canterbury's Craigieburn Range,

which is also east of the Main Divide, Prowse and Owens (1982) found that sensible heat dominated melt energetics, which suggests that spatial gradients in climatic sensitivity are not uniform across the South Island. Further complicating the relative importance of turbulent and radiative fluxes in controlling ablation processes is the role of wind entrainment. As discussed previously, wind re-distribution of snow is likely to result in concentration of snow as promoted by topography, and can promote ablation of the snow pack by sublimation under favourable conditions (Pomeroy and Gray, 1990; Pomeroy and Essery, 1999;

Wang and Jia, 2018). Preferentially accumulated snow will generally increase in density and depth, further increasing the energy required to ablate the snow pack. The increased rate at which mean SCD lengthens with elevation on southerly aspects compared to northerly aspects further supports this interpretation.

Spatial variability in the sensitivity of seasonal snow to climatic parameters has been detected elsewhere. Howat and Tulaczyk (2005), for example, detected spatial variability in sensitivity of maximum spring SWE to climate forcings in the

predominantly precipitation sensitive Sierra Nevada of California. While a portion of observed sensitivity was attributed to elevation, an unexplained latitudinal dependence was attributed to contrasting regional atmospheric circulation patterns. As pointed out by Howat and Tulaczyk (2005), observations of spatial variability in temperature sensitivity may lead to over-predictions of reduced snow cover by models that lack sufficient spatial scale or parameterisation. These findings can provide guidance for efforts to model seasonal snow processes at the catchment to national scale in the New Zealand context. It is

clear, for example, that while a degree-day model may perform adequately at recreating melt on and near the main divide, its performance will likely deteriorate for ranges further east, where the relative importance of variability in energy balance and wind-driven redistribution of snow is increased. Hock (2003) emphasised that while degree-day models work well for modelling melt at catchment scales over long time periods, reliability is reduced as temporal resolution increases and where spatial variability in melt rates exist. Similarly, Clark et al. (2009) suggested that basin specific parameterisations are required

to improve the simulation of seasonal snow across the South Island, and these results can provide guidance to this end within the Clutha Catchment, as well as providing a framework for the assessment of snow cover sensitivity to climatic forcings that could be scaled up across the South Island.

There is a need to better understand the relative importance of temperature and precipitation sensitivity, in particular to better capture the potential for gains in snow accumulation through increased precipitation to offset losses driven by increased

temperature. Furthermore, the current study indicates that considering just temperature and precipitation leaves a large com-



ponent of SCD variability unexplained, emphasising the need to improve understanding and parameterisation of individual components of the energy balance and redistribution of snow by wind. Degree day models may be enhanced relatively simply through the inclusion of radiation and albedo factors. Capturing the effects of wind on snow redistribution, however, remains a substantial challenge, further compounded by a scarcity of data in alpine environments such as New Zealand (Hendrikx and

Harper, 2013), and the need to characterise terrain at sufficient resolution to resolve wind-driven processes at relevant scales (e.g., Marsh et al., 2018; Mott et al., 2018).

### 5.4.3    Atmospheric circulation

The association between HYSPLIT trajectory anomalies and characteristic spatial variability in SCD, as expressed by PC loadings, provides new insight into the role of anomalous synoptic scale circulation on snow cover variability. For hy 2011

in particular, relatively high positive loadings were identified for PC2 & 3, which are both associated with positive anomalies in SCD in the eastern part of the catchment (particularly the Manuherikia basin), and in the case of PC2, relatively strong positive anomalies in SCD at low elevations across the catchment. HYSPLIT trajectory anomalies for this period revealed a substantial increase in the frequency of airflow originating to the east of New Zealand. This scenario of anomalous easterly airflow represents a reversal of the typical zonal flow affecting the South Island of New Zealand, which is typically dominated

by westerly flow (Sturman and Tapper, 2006; Macara, 2015). The 2011 hydrological year also featured the highest observed mean winter SOI value (i.e., the strongest observed La Niña event).

Overall, the climatic indices of SOI and SAM did not provide strong explanations for spatio-temporal variability of snow cover within the Clutha Catchment. A notable exception to this was hy 2012. This year had the largest positive loading on PC2, corresponding to an annular spatial pattern in SCD anomaly with positive anomalies at low elevations and negative anomalies at

high elevations. Climatically, this hy was characterised by strong positive SOI values (i.e., La Niña conditions) from February 2011, weakening somewhat by June, before strengthening again through to December (Renwick, 2012). At the same time, the SAM switched from positive in May to near-neutral in June, before remaining strongly negative from July to September. Subsequently, this snow season was substantially shortened at high elevations, due to a reduction in accumulation early in winter. Conversely, the persistently negative SAM through late winter was accompanied by a period of regular low level snow

fall that extended SCD at low elevations.

The most reduced SCD across the catchment occurred for hys 2006 and 2015, which were also the most similar in terms of PC loadings. In particular, PC1 was associated with positive anomalies in airflow originating north-east of New Zealand. While positive SOI values (e.g., La Niña conditions) are associated with anomalous north-easterly flow for New Zealand (Sturman and Tapper, 2006), these years were associated with both slightly positive (hy 2006) and negative (hy 2015) winter mean SOI

values, while both having positive anomalies in the frequency of north-easterly airflow. This analysis highlights that, despite the complex interacting processes governing seasonal snow cover in New Zealand, large scale climatic perturbations can be linked to the spatio-temporal distribution of seasonal snow cover, beyond controls imposed by regional topography. An expected future strengthening of winter westerlies for the lower South Island (Mullan et al., 2016) would decrease the likelihood of



extended SCD in the eastern part of the Clutha Catchment, through a corresponding reduction in winter easterly flow, which is important for delivering precipitation to the Manuherikia basin.

## 5.5 Limitations of scale

The large number of principal components required to explain the observed variability suggest that some of the spatial variability affecting seasonal snow within the Clutha Catchment occurs beyond the detection limits of MODIS. While an inability to resolve fine scale processes limits insights into the full range of processes controlling seasonal snow within the catchment, spatial variability in sensitivity of SCD to changes in temperature and precipitation has been demonstrated at a sub-catchment scale. Furthermore, the observed degradation of the relationship between SCD and elevation away from the Main Divide (Figure 13) suggests the processes occurring at scales that are difficult to detect with MODIS gain significance in controlling SCD. These processes include re-distribution of snow by wind and the associated influence of micro-terrain, which may complicate the interpretation of pixel level fSCA. This is likely to be especially relevant to hydrological modelling and demands further attention on fine-scale spatial variability in seasonal snow.

## 6 Conclusions

Conclusions and outlook This study has produced a 16-year time series of daily snow covered area from MODIS imagery for the Clutha Catchment, New Zealand, providing an important supplement to sparse seasonal snow observations in the Southern Hemisphere. From the daily SCA time series, a number of metrics have been derived to characterise the spatio-temporal variability of seasonal snow cover for this region. A consistent reduction in mid-winter SCA demonstrates the influence of persistent synoptic scale features, such as blocking highs. Overall, the inter-annual variability is high, concealing any trend that may exist over the study period. Despite the lack of a temporal trend in this record, extreme years provide insight into conditions that may become more frequent under future climate change scenarios.

Characterisation of spatial modes of variability via spatialised rPCA reinforces the highly dynamic nature of seasonal snow cover within the catchment, with minimal similarity in spatial patterns between years. Nonetheless, this analysis proved capable of identifying the two most similar years, however, for which spatial patterns of SCD were linked to anomalous north-easterly airflow. The ability to identify such associations highlights the strength of the approach taken to rPCA, which retained both the spatial and temporal signals of SCD variability. Spatial variability in snow cover across the catchment is also high, with the SCD anomaly being out of phase for some years, indicating that the Clutha Catchment departs markedly from a single climatic unit in terms of seasonal snow.

Spatial variability in the sensitivity of SCD to temperature and precipitation variability at sub-catchment scale has also been demonstrated. Temperature was found to be an important control of SCD variability on and near the main divide. Precipitation was important in the east of the Main Divide, but insignificant for ranges further east. On their own, these two variables leave a large proportion of SCD variability unexplained. While providing guidance for efforts to improve parameterisations of snow models in New Zealand, these findings also highlight the likely importance of other components of the energy balance and the





redistribution of snow by wind in influencing seasonal snow processes, and subsequent shortcomings in traditional degree-day approaches to modelling seasonal snow in this region. In demonstrating the utility of rPCA for examining spatio-temporal variability in SCD at regional scales, this work provides a framework for the ongoing monitoring of seasonal snow cover, improved context for understanding the snow hydrology of the Clutha Catchment, and can contribute to efforts to improve the

5 modelling of seasonal snow throughout New Zealand and elsewhere, particularly where in situ records are sparse.

*Code and data availability.*  MODIS data used in this research are freely available from the Level 1 and Atmosphere Archive and Distribution System (LAADS) Web. The MODImLab software is available on request from P. Sirguey.

*Author contributions.*  T. Redpath, P. Sirguey and N. Cullen designed the study. T. Redpath and P. Sirguey processed and analysed the data. P. Sirguey developed and maintains the MODImLab software. T. Redpath prepared the manuscript with contributions from all authors.

10 *Competing interests.*  The authors declare they have no conflicts of interest.

*Acknowledgements.*  We are grateful to Andreas Jobst for providing gridded temperature and precipitation data for the Clutha Catchment. Sean Fitzsimons provided helpful feedback on the draft manuscript. T. Redpath was funded by a University of Otago Doctoral Scholarship.



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





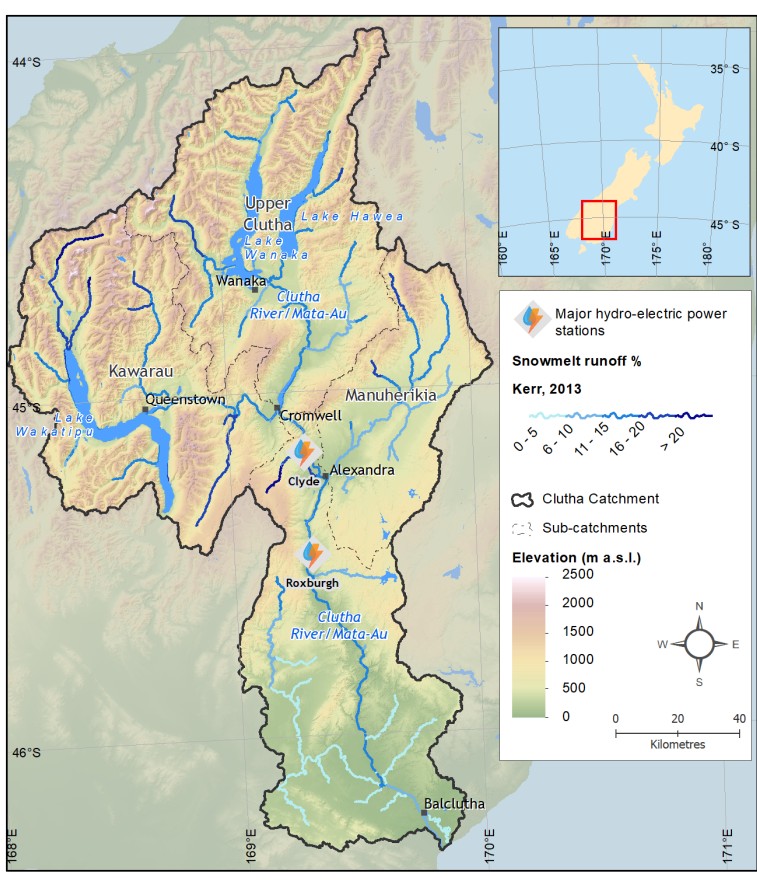

**Figure 1.** Context map of the Clutha Catchment, including snow melt contribution to stream flow for rivers of stream order 4 and greater, snow melt contribution data from Kerr (2013). The boundaries of the Upper Clutha, Kawarau and Manuherikia sub-catchments are also included.





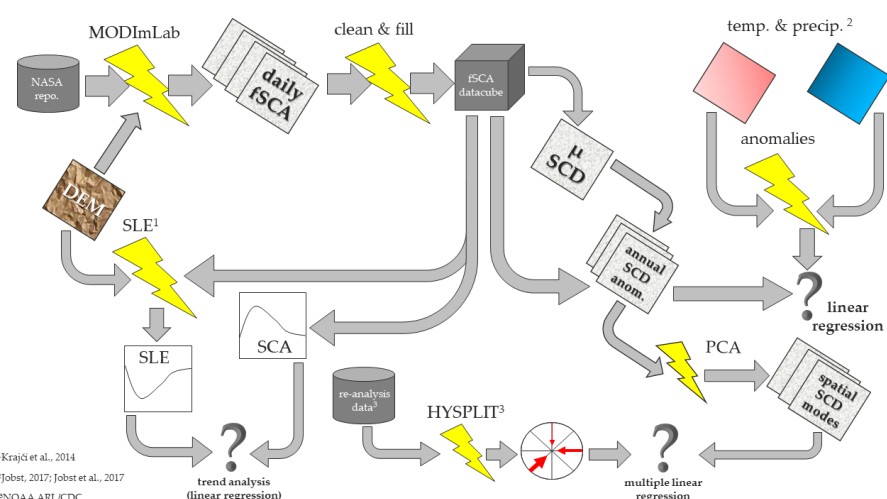

**Figure 2.** MODIS imagery processing chain and subsequent analytical steps applied. The fractional snow covered area (fSCA) datacube facilitates the determination of daily snowline elevation (SLE) and snow covered area (SCA). The mean snow cover duration ($\mu$SCD) calculated over the length of the time series provides the basis for calculating annual SCD anomalies. Along with spatial SCD modes from PCA analysis these form the basis for assessing the importance of climatic forcings on spatio-temporal variability in SCD.





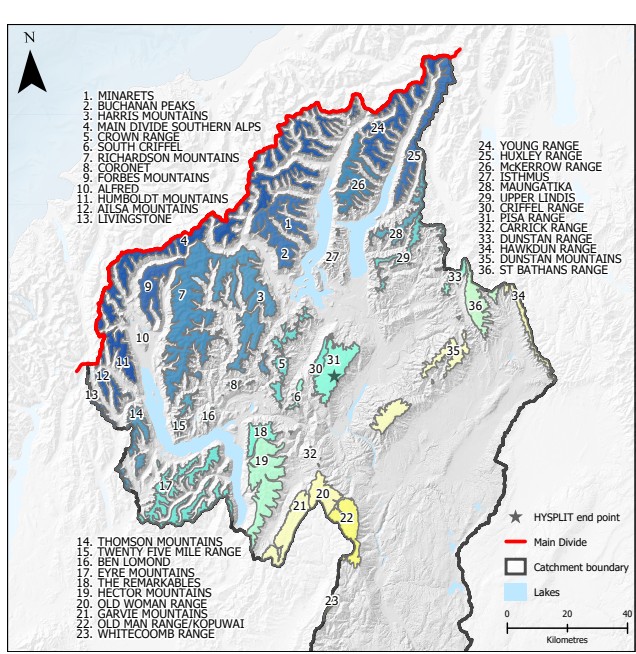

**Figure 3.** Individual mountain ranges mapped out within the Clutha Catchment for analysis of SCD variability and temperature and precipitation sensitivity. The end point for HYSPLIT back trajectories is also shown.



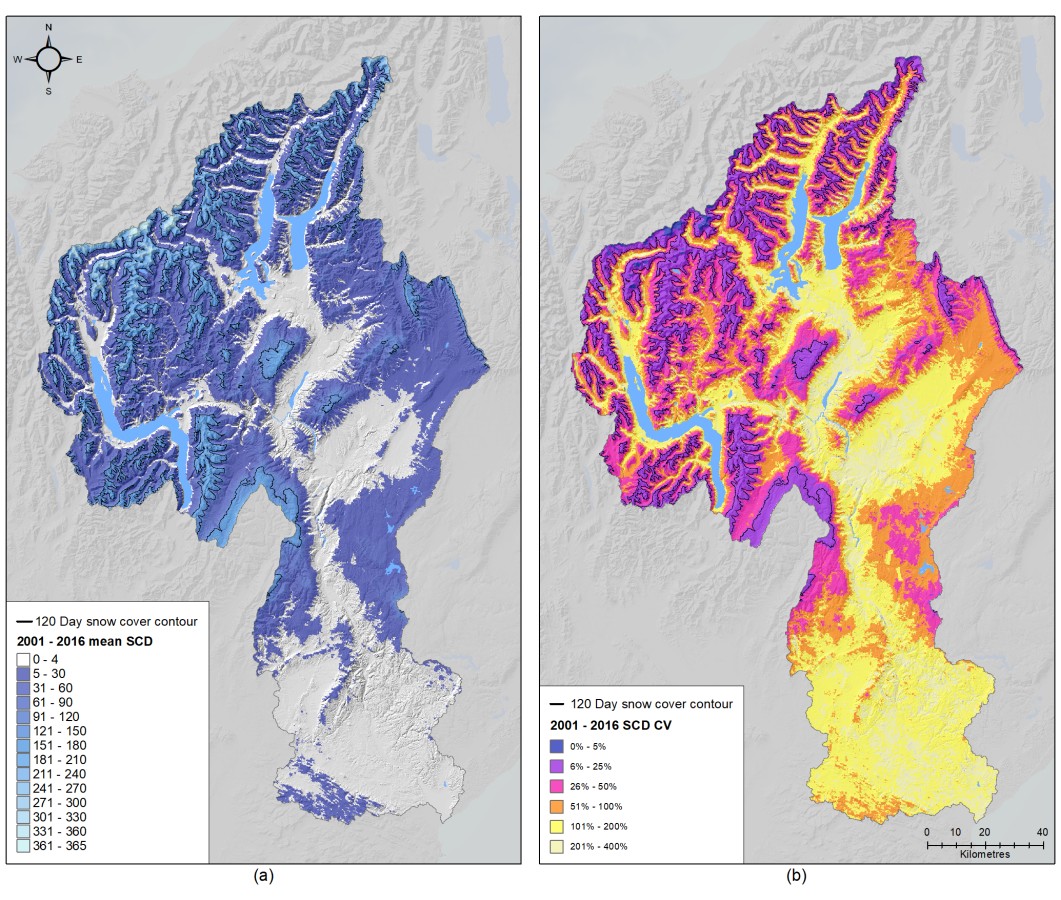

**Figure 4.** Mean annual SCD,( $\mu_{SCD}$ ), (a) and associated coefficient of variation, CV ($CV_{SCD}$), (b) for the Clutha Catchment, 2001-2016.





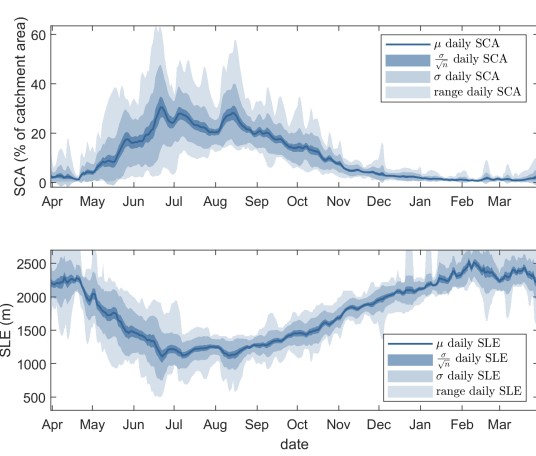

**Figure 5.** 10-day moving average of daily snow covered area (SCA) (top) and snowline elevation (SLE) (bottom). The mean of each is shown in black, accompanied by the envelopes of associated standard deviation, range and standard error to illustrate the spread of trajectories, and the uncertainty of the mean, respectively.





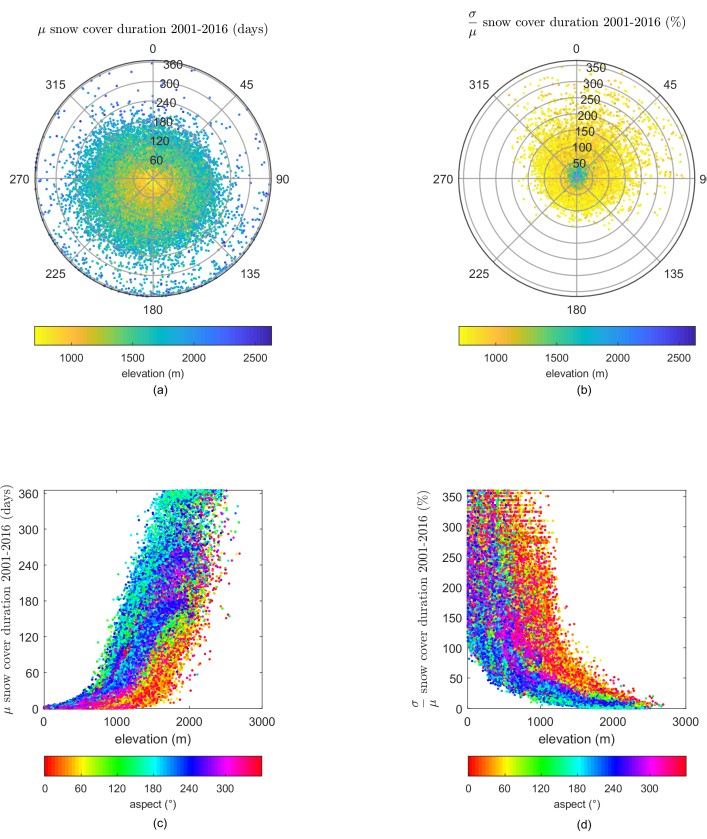

**Figure 6.** Mean SCD (left) and associated coefficient of variability ($\frac{\sigma}{\mu}$) (right) as a function of aspect (top), and elevation (bottom). For clarity, a random sub-sample of 50,000 pixels is displayed in the plots.

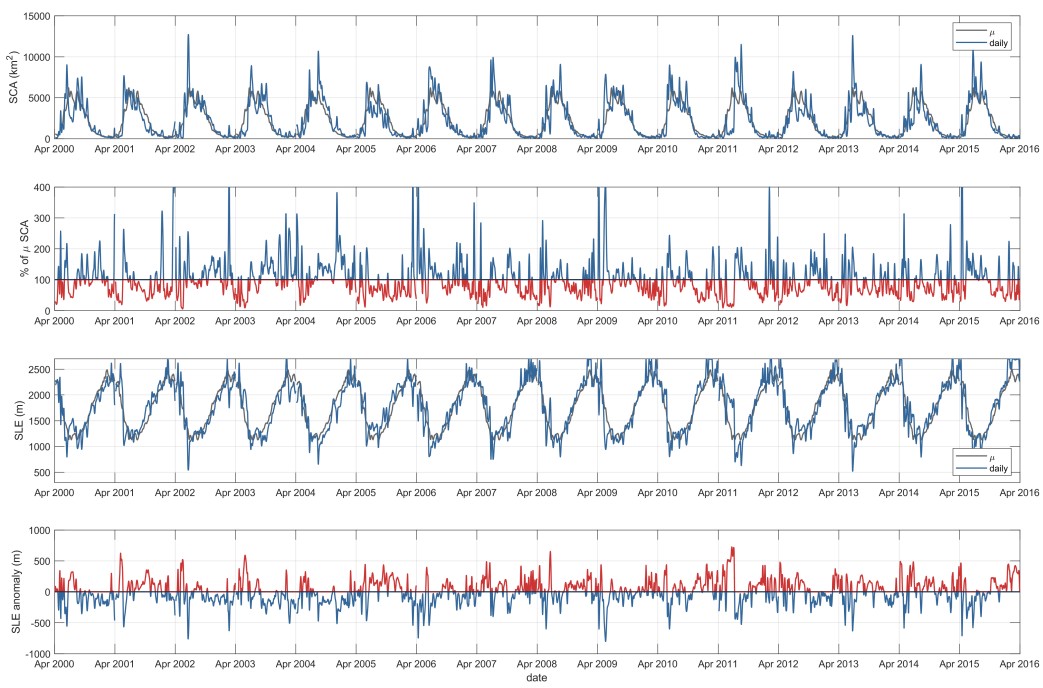

**Figure 7.** 5-day moving average of daily snow covered area (SCA) and percentage of mean snow covered area for all years (upper two panels), and daily snowline elevation (SLE) and daily SLE anomaly (lower two panels). Mean daily values, repeated for each year, are shown in gray in the daily SCA and SLE plots.





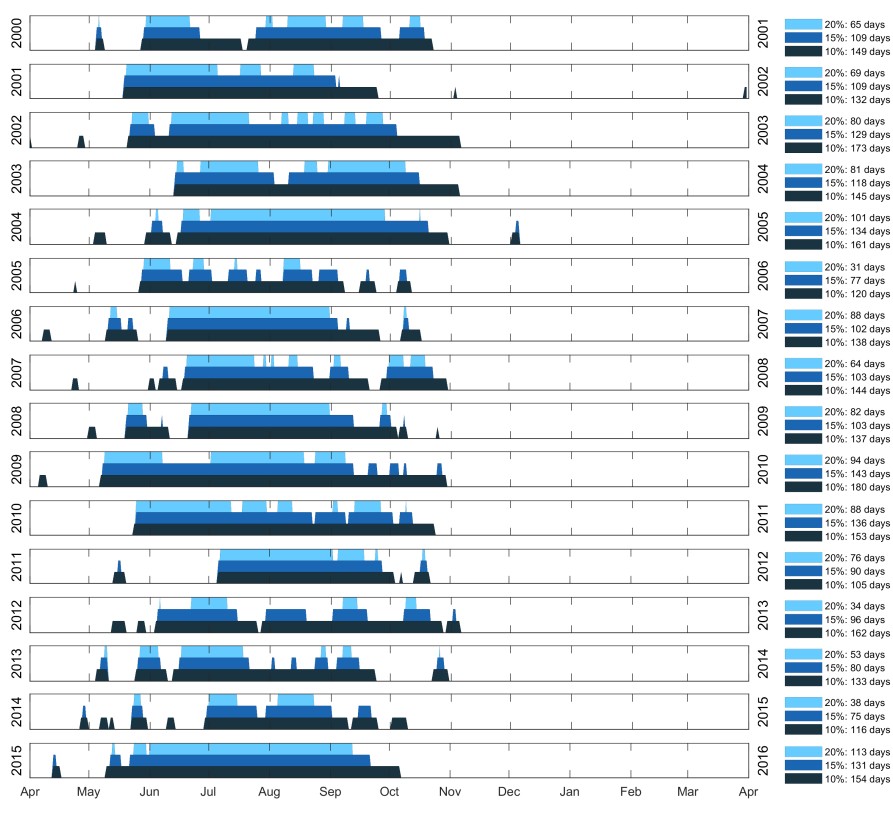

**Figure 8.** Timeline of occurrence of 10, 15 and 20% basin snow covered area (bSCA) for the Clutha Catchment for hydrological years 2001-2016.



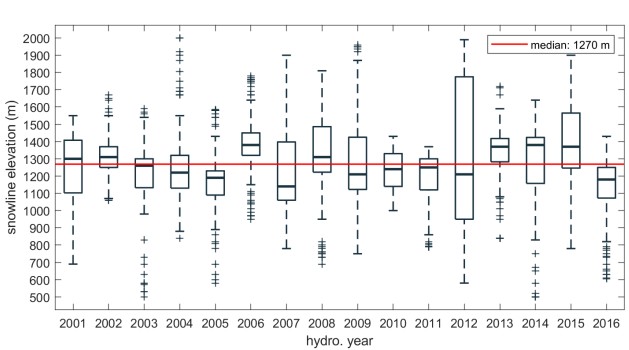

**Figure 9.** Box-plots of winter (01 June-30 September) snowline elevation for all years. The upper and lower bounds of the boxes are the 75th and 25th percentiles, respectively. Outliers are represented by '+' symbols.





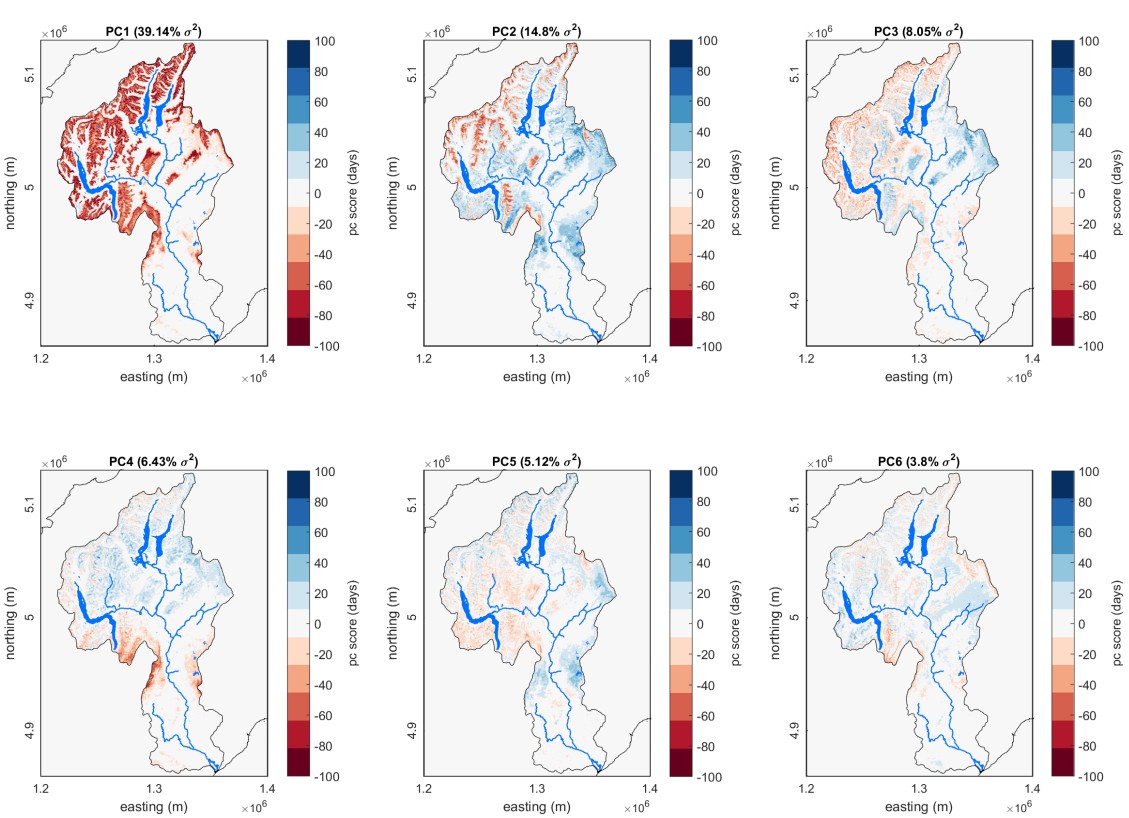

**Figure 10.** The first six spatial principal components of SCD anomaly.

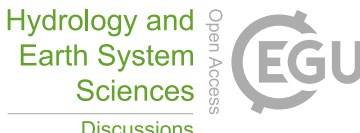



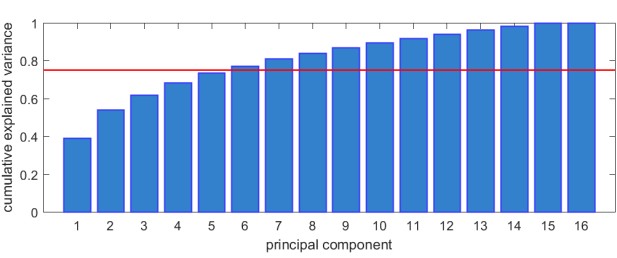

**Figure 11.** Cumulative explained variance for 16 principal components of SCD anomaly. The red line indicates 77% of explained variance.





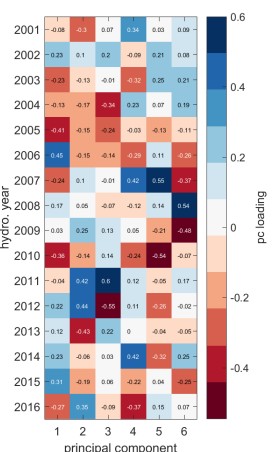 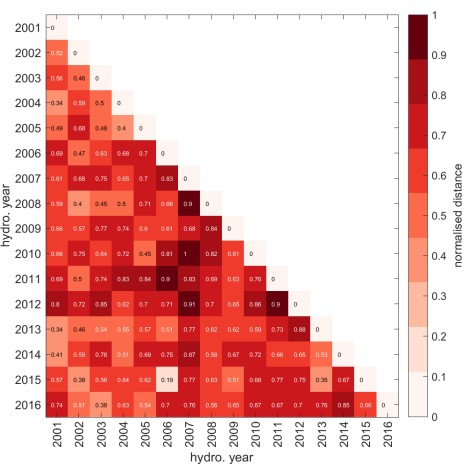

**Figure 12.** (Left) Loadings for the first 6 principal components of SCD anomaly (77% of total variance) for each hydrological year (which corresponds with the winter of the preceding calendar year). (Right) Pair-wise distance between PC loadings.

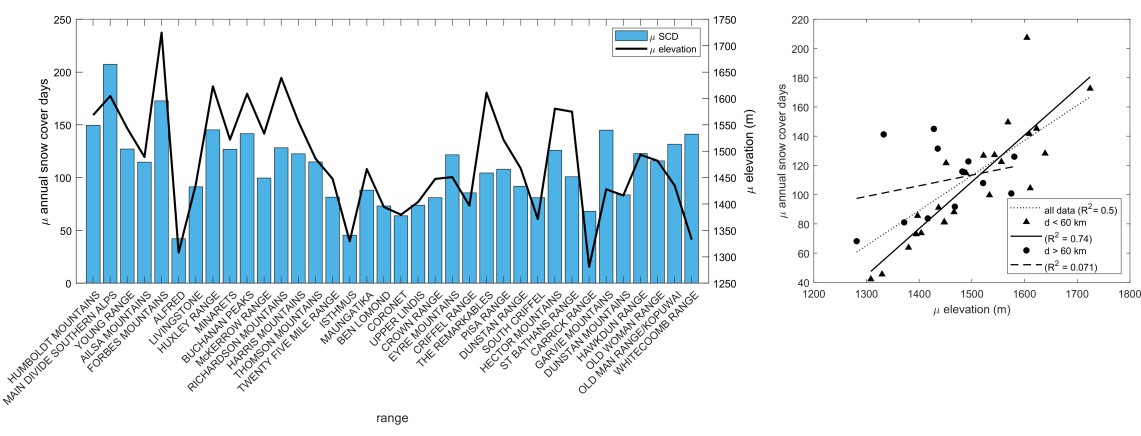

**Figure 13.** Relationship between mean annual SCD, and mean elevation of each mountain range. Ranges are sorted from left to right by their proximity to the main divide of the Southern Alps. The relationship between elevation and snow cover persistence was found to be much stronger in the western (within 60 km of the main divide) part of the catchment than in the east.




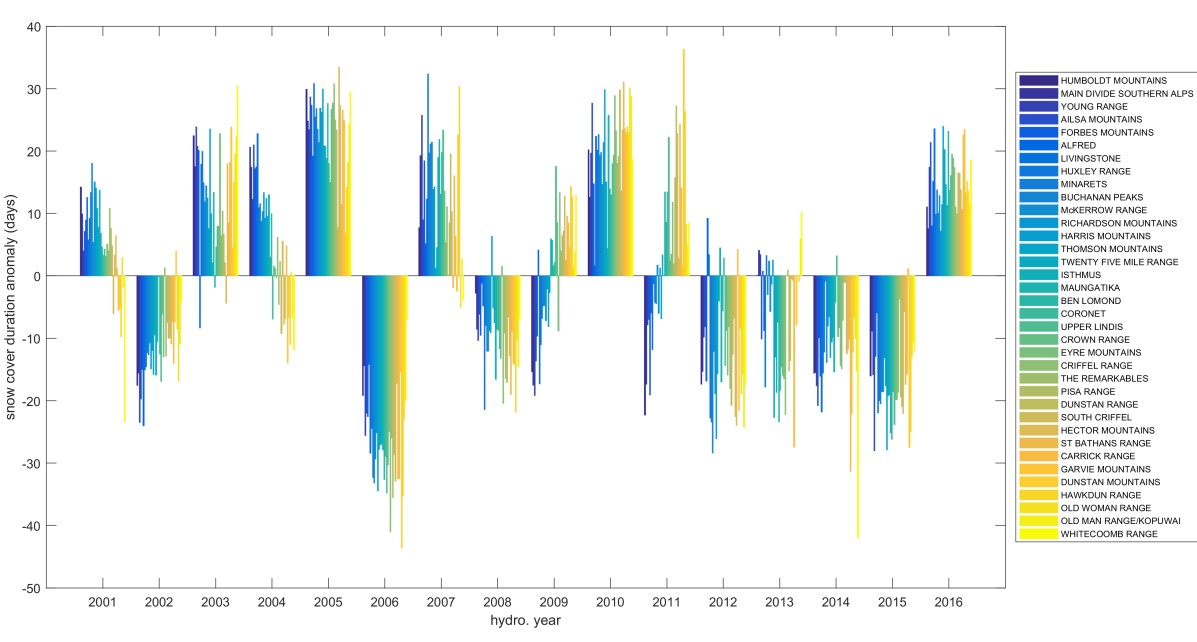

**Figure 14.** SCD anomaly (in days) for geographic units of mountain ranges within the Clutha Catchment. Mountain ranges are sorted and symbolised by their proximity to the main divide of the Southern Alps, revealing spatial gradients in SCD anomaly across the catchment.



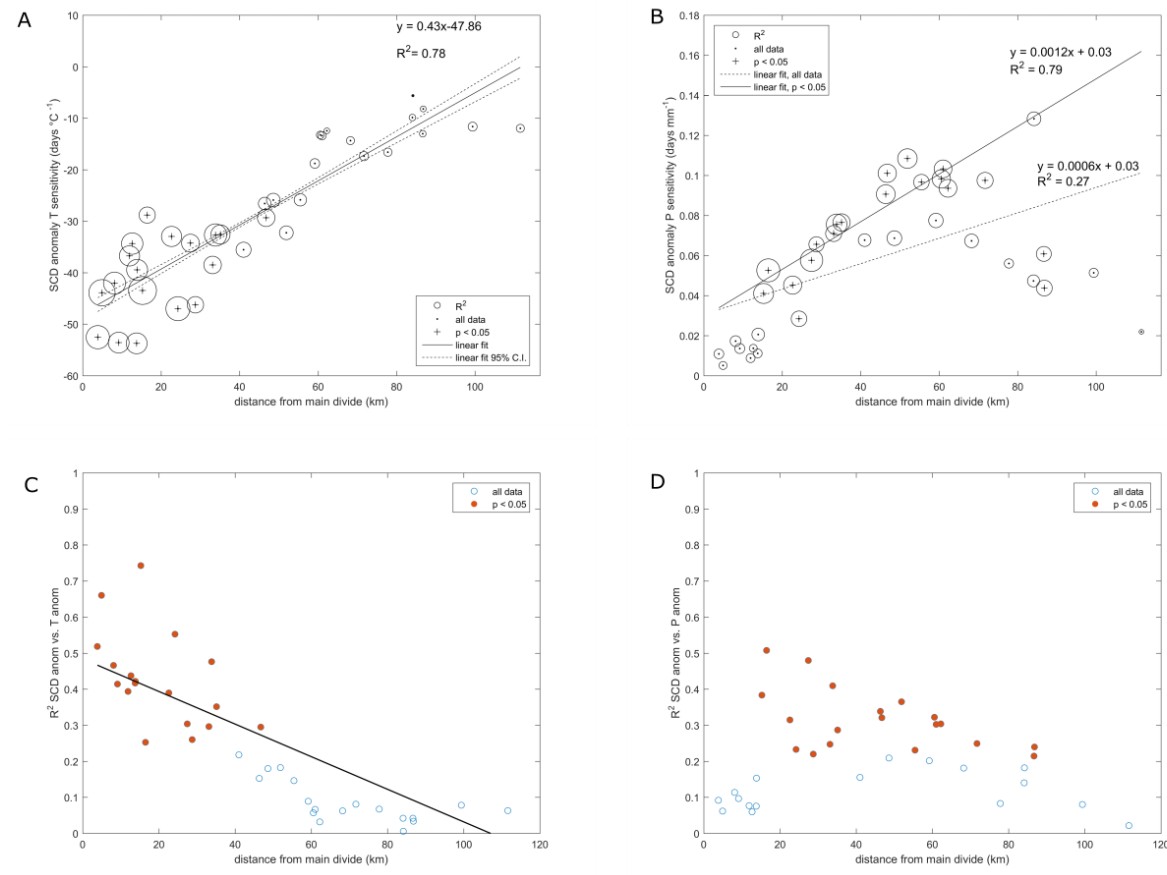

**Figure 15.** Regressions between the sensitivity of SCD to temperature (A) and precipitation (B) and correlation coefficient ($R^2$) of the relationship between range SCD and temperature (C) and precipitation (D) in terms of distance of each range from the Main Divide.





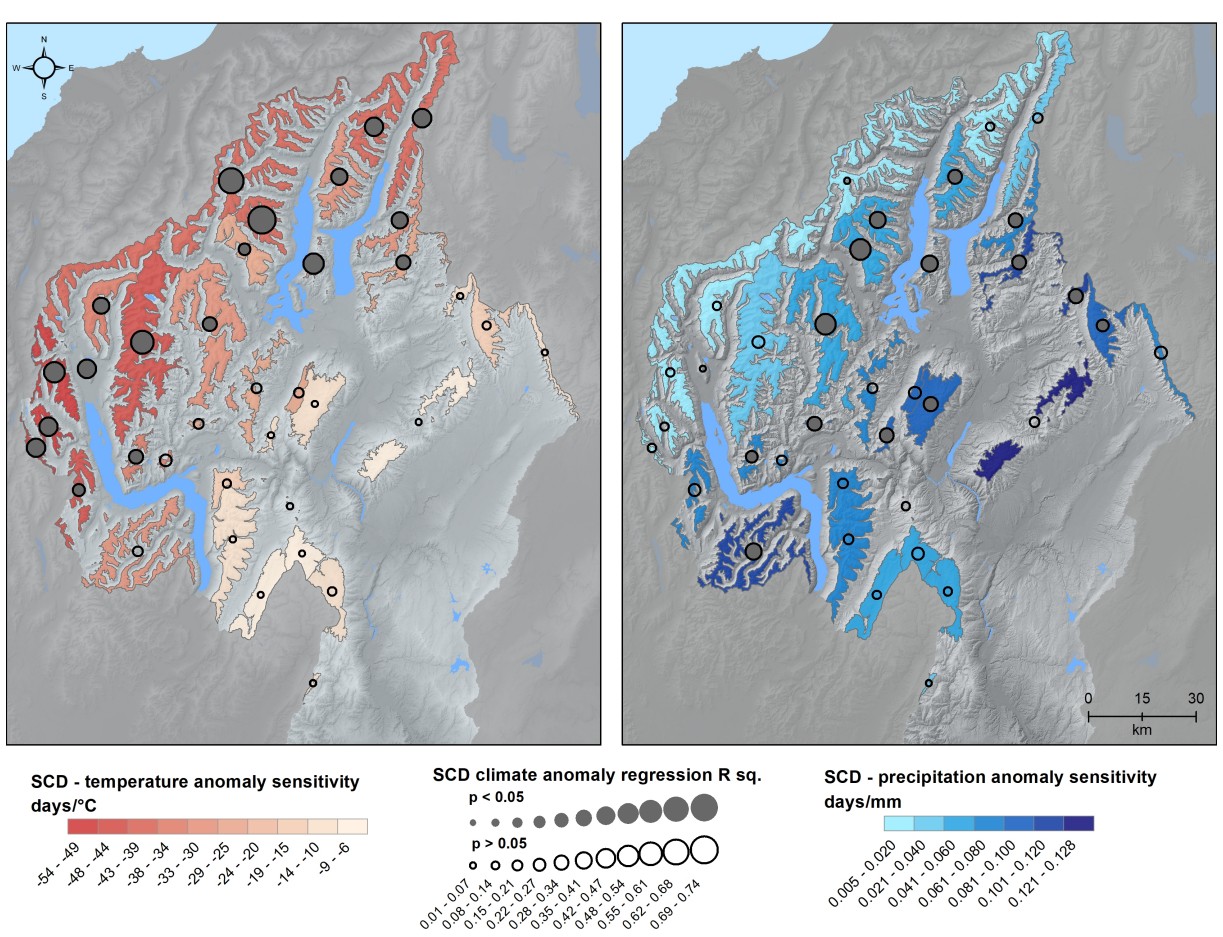

**Figure 16.** Spatial variability in the regression parameters of slope and $R^2$ between SCD anomaly and temperature and precipitation anomalies for each mountain range within the Clutha Catchment.





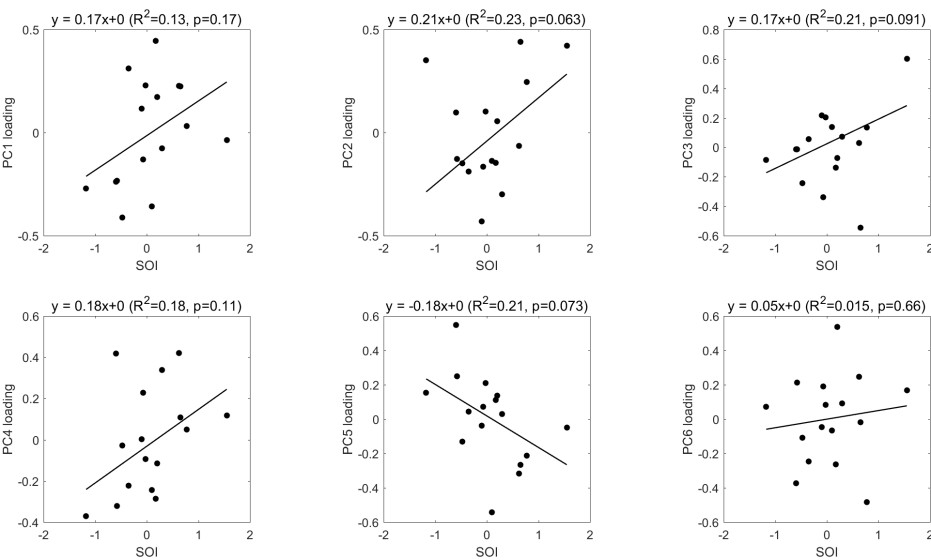

**Figure 17.** Scatter plots with linear regression between average winter SOI phase and annualised loadings for principal components 1-6.



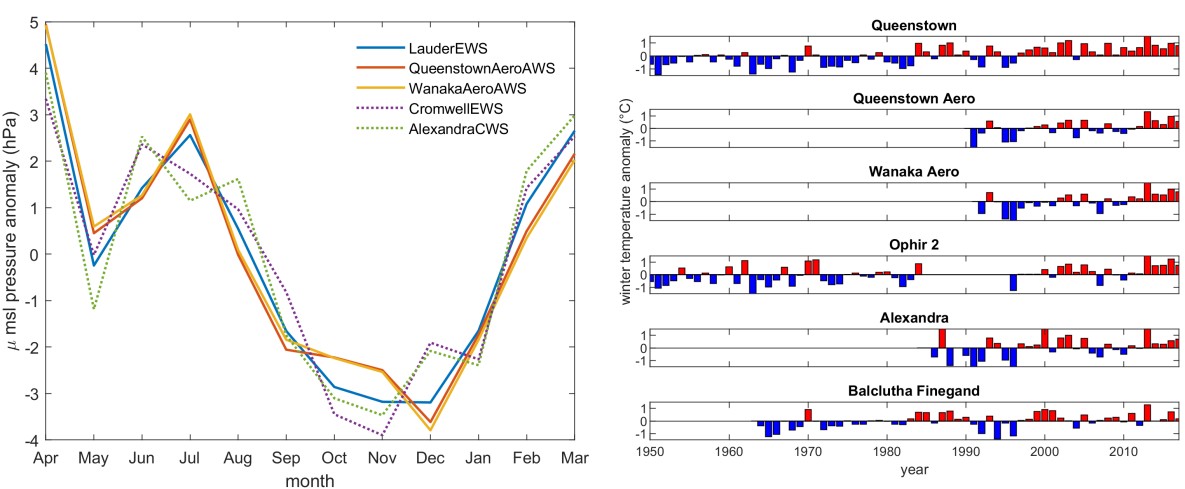

**Figure 18.** (Left) Mean monthly atmospheric pressure (mean sea level) plotted for five climate stations within the Clutha Catchment. Record lengths vary from eight (dashed lines) to 16 years (solid lines). (Right) Anomalies in winter mean air temperature for low elevation stations in the Clutha Catchment. Anomalies for each station were calculated with respect to the mean of the the station record. Data from the NIWA National Climate Database(CliFlo) (NIWA, 2018).



**Table 1.** Landcover classes above 1000 m in the Clutha Catchment, data from the New Zealand Landcover Database v4.1 (Landcare Research, 2015).

| Class | Area (km$^2$) | Class % |
|---|---|---|
| All other classes | 108.8 | 1.7 |
| Permanent Snow and Ice | 144.6 | 2.2 |
| Indigenous Forest | 185.0 | 2.8 |
| Low Producing Grassland | 225.0 | 3.4 |
| Alpine Grass/Herbfield | 269.6 | 4.1 |
| Sub Alpine Shrubland | 320.4 | 4.9 |
| Gravel or Rock | 811.5 | 12.3 |
| Tall Tussock Grassland | 4537.7 | 68.7 |
| Total | 6602.4 | 100.0 |





**Table 2.** Basin wide snow covered area (bSCA) temporal metrics. These metrics were calculated annually, per-pixel.

| Name | Description | bSCA (%) |
|---|---|---|
| tSCD | Total number of snow cover days | 10, 15, 20 |
| maxSCD | Maximum continuous number of days exceeding $fSCA_t$ | 10, 15, 20 |
| $maxSCD_{onset}$ | First day of maxSCD | 10, 15, 20 |
| $maxSCD_{loss}$ | Last day of maxSCD | 10, 15, 20 |

**Table 3.** k-means clusters based on principal component loadings of SCD anomalies.

| Cluster | | | |
|---|---|---|---|
| 1 | 2 | 3 | 4 |
| 2007 | 2002 | 2001 | 2003 |
| 2009 | 2006 | 2004 | 2005 |
| 2011 | 2013 | 2008 | 2010 |
| - | 2015 | 2012 | 2016 |
| - | - | 2014 | - |





**Table 4.** Robust (bi-square) regression parameters between mean elevation and mean SCD for mountain ranges within the Clutha Catchment. Regression was carried out for all ranges together, and for those ranges within 60 km of the main divide, and more than 60 km from the main divide.

| Ranges | $n$ | $\beta$ (d m$^{-1}$) | $c$ | $R^2$ | RMSE | $p$ |
|---|---|---|---|---|---|---|
| All | 36 | 0.24 | $-246.11$ | 0.50 | 24.8 | 0.00 |
| < 60 km of M.D. | 24 | 0.32 | $-370.52$ | 0.74 | 20.3 | 0.00 |
| > 60 km of M.D. | 12 | 0.07 | 3.89 | 0.07 | 25.1 | 0.41 |





**Table 5.** Parameters for robust (bi-square) linear regressions between snow cover anomaly and winter temperature/precipitation anomalies for individual ranges and all ranges together. Ranges are listed in the table by their proximity to the main divide.

| Name | Temperature | | | | | Precipitation | | | | |
|------|-------------|---|---|---|------|---------------|---|---|---|------|
| | $\beta$ (d °C$^{-1}$) | $c$ | R$^2$ | $p$ | RMSE | $\beta$ (d mm$^{-1}$) | $c$ | R$^2$ | $p$ | RMSE |
| HUMBOLDT MOUNTAINS | -52.54 | 2.52 | 0.52 | 0.00 | 13.91 | 0.01 | 0.02 | 0.09 | 0.25 | 19.33 |
| MAIN DIVIDE SOUTHERN ALPS | -43.88 | 2.06 | 0.66 | 0.00 | 9.85 | 0.01 | -0.14 | 0.06 | 0.35 | 16.58 |
| YOUNG RANGE | -42.02 | 0.77 | 0.47 | 0.00 | 14.90 | 0.02 | -0.14 | 0.11 | 0.20 | 19.17 |
| AILSA MOUNTAINS | -53.59 | 1.18 | 0.41 | 0.01 | 15.68 | 0.01 | 0.16 | 0.10 | 0.24 | 19.44 |
| FORBES MOUNTAINS | -36.72 | 0.72 | 0.39 | 0.01 | 13.53 | 0.01 | -0.17 | 0.08 | 0.30 | 16.76 |
| ALFRED | -34.37 | -0.30 | 0.44 | 0.01 | 10.87 | 0.01 | -0.02 | 0.06 | 0.36 | 14.01 |
| LIVINGSTONE | -53.72 | 0.42 | 0.42 | 0.01 | 15.34 | 0.01 | -0.01 | 0.08 | 0.30 | 19.40 |
| HUXLEY RANGE | -39.46 | 0.03 | 0.42 | 0.01 | 15.58 | 0.02 | -0.64 | 0.15 | 0.14 | 19.24 |
| MINARETS | -43.48 | 3.82 | 0.74 | 0.00 | 9.19 | 0.04 | -0.24 | 0.38 | 0.01 | 15.86 |
| BUCHANAN PEAKS | -28.80 | 0.88 | 0.25 | 0.05 | 16.97 | 0.05 | -0.32 | 0.51 | 0.00 | 14.02 |
| McKERROW RANGE | -32.98 | 0.40 | 0.39 | 0.01 | 14.17 | 0.05 | -0.36 | 0.31 | 0.02 | 15.05 |
| RICHARDSON MOUNTAINS | -46.98 | 3.23 | 0.55 | 0.00 | 12.42 | 0.03 | -0.25 | 0.23 | 0.06 | 15.48 |
| HARRIS MOUNTAINS | -34.25 | 1.66 | 0.30 | 0.03 | 16.04 | 0.06 | 0.00 | 0.48 | 0.00 | 13.46 |
| THOMSON MOUNTAINS | -46.21 | 0.06 | 0.26 | 0.04 | 18.49 | 0.07 | 0.47 | 0.22 | 0.07 | 20.03 |
| TWENTY FIVE MILE RANGE | -38.50 | -0.27 | 0.30 | 0.03 | 14.54 | 0.07 | 0.05 | 0.25 | 0.05 | 15.37 |
| ISTHMUS | -32.68 | -0.53 | 0.48 | 0.00 | 12.33 | 0.08 | -0.24 | 0.41 | 0.01 | 13.60 |
| MAUNGATIKA | -32.57 | -0.86 | 0.35 | 0.02 | 15.96 | 0.08 | -0.35 | 0.29 | 0.03 | 16.49 |
| BEN LOMOND | -35.53 | 0.57 | 0.22 | 0.07 | 16.96 | 0.07 | 0.10 | 0.15 | 0.13 | 18.48 |
| CORONET | -26.59 | 0.03 | 0.15 | 0.13 | 16.22 | 0.09 | 0.08 | 0.34 | 0.02 | 14.06 |
| UPPER LINDIS | -29.32 | -0.95 | 0.30 | 0.03 | 16.64 | 0.10 | -0.08 | 0.32 | 0.02 | 16.32 |
| CROWN RANGE | -25.91 | -0.23 | 0.18 | 0.10 | 16.16 | 0.07 | -0.20 | 0.21 | 0.08 | 15.61 |
| EYRE MOUNTAINS | -32.26 | -0.25 | 0.18 | 0.10 | 17.25 | 0.11 | -0.15 | 0.37 | 0.01 | 15.83 |
| CRIFFEL RANGE | -25.85 | -0.27 | 0.15 | 0.14 | 19.62 | 0.10 | -0.25 | 0.23 | 0.06 | 17.97 |
| THE REMARKABLES | -18.81 | 0.37 | 0.09 | 0.26 | 16.35 | 0.08 | 0.04 | 0.20 | 0.08 | 15.38 |
| PISA RANGE | -13.28 | 0.26 | 0.06 | 0.38 | 17.53 | 0.10 | -0.08 | 0.32 | 0.02 | 14.74 |
| DUNSTAN RANGE | -13.41 | -0.67 | 0.07 | 0.34 | 18.34 | 0.10 | -0.81 | 0.30 | 0.03 | 15.03 |
| SOUTH CRIFFEL | -12.44 | -0.98 | 0.03 | 0.52 | 21.11 | 0.09 | -0.41 | 0.30 | 0.03 | 16.60 |
| HECTOR MOUNTAINS | -14.37 | -0.07 | 0.06 | 0.35 | 16.90 | 0.07 | 0.01 | 0.18 | 0.10 | 15.58 |
| ST BATHANS RANGE | -17.36 | -0.22 | 0.08 | 0.29 | 20.45 | 0.10 | -0.45 | 0.25 | 0.05 | 16.71 |
| CARRICK RANGE | -16.59 | 0.02 | 0.07 | 0.33 | 21.34 | 0.06 | 0.48 | 0.08 | 0.28 | 20.44 |
| GARVIE MOUNTAINS | -9.86 | -0.36 | 0.04 | 0.45 | 17.33 | 0.05 | -0.08 | 0.14 | 0.15 | 16.19 |
| DUNSTAN MOUNTAINS | -5.63 | 0.35 | 0.01 | 0.78 | 24.06 | 0.13 | -0.33 | 0.18 | 0.10 | 20.76 |
| HAWKDUN RANGE | -13.01 | -0.08 | 0.04 | 0.45 | 20.56 | 0.06 | -0.22 | 0.21 | 0.07 | 18.38 |
| OLD WOMAN RANGE | -8.20 | -0.31 | 0.03 | 0.50 | 17.27 | 0.04 | -0.50 | 0.24 | 0.05 | 14.08 |
| OLD MAN RANGE/KOPUWAI | -11.66 | -0.38 | 0.08 | 0.30 | 17.19 | 0.05 | -0.83 | 0.08 | 0.30 | 17.64 |
| WHITECOOMB RANGE | -11.96 | 0.08 | 0.06 | 0.35 | 19.26 | 0.02 | 0.45 | 0.02 | 0.60 | 19.35 |
| All | -25.50 | -0.07 | 0.19 | 0.00 | 15.91 | 0.02 | 0.07 | 0.11 | 0.00 | 16.76 |



**Table 6.** Parameters of regression between basin snow cover metrics and winter SOI and SAM values.

| Metric | bSCA (%) | SOI | | | SAM | | |
|---|---|---|---|---|---|---|---|
| | | sign $\beta$ | $R^2$ | $p$ | sign $\beta$ | $R^2$ | $p$ |
| tSCD | 10 | - | 0.06 | 0.37 | + | 0.04 | 0.46 |
| | 15 | - | 0.02 | 0.62 | + | 0.14 | 0.16 |
| | 20 | - | 0.04 | 0.46 | + | 0.10 | 0.23 |
| maxSCD | 10 | - | 0.02 | 0.56 | + | 0.07 | 0.31 |
| | 15 | - | 0.08 | 0.29 | + | 0.05 | 0.37 |
| | 20 | - | 0.11 | 0.20 | + | 0.17 | 0.10 |
| maxSCD$_{onset}$ | 10 | + | 0.03 | 0.51 | - | 0.03 | 0.50 |
| | 15 | + | 0.00 | 0.95 | - | 0.07 | 0.33 |
| | 20 | + | 0.02 | 0.63 | - | 0.03 | 0.52 |
| maxSCD$_{loss}$ | 10 | - | 0.00 | 0.91 | + | 0.04 | 0.46 |
| | 15 | - | 0.08 | 0.30 | + | 0.00 | 0.89 |
| | 20 | - | 0.12 | 0.18 | + | 0.03 | 0.50 |





**Table 7.** Parameters for significant ($p \leq 0.05$) linear multiple regression models between PC loadings and north-east (NE), east (E), south-east (SE) and south (S) trajectory frequency anomalies over varying time periods. Significant ($p \leq 0.05$) predictive variables are denoted by *.

| PC | t (h) | $\beta_0$ | $\beta_1$ (NE) | $\beta_2$ (E) | $\beta_3$ (SE) | $\beta_4$ (S) | $R^2$ | adj. $R^2$ | $p$ |
|---|---|---|---|---|---|---|---|---|---|
| 1 | -96 | -0.21 | 0.41* | 0.00 | -0.03 | -0.19 | 0.59 | 0.44 | 0.03 |
| 2 | -72 | -0.28 | -0.23 | -0.34* | 0.29* | 0.52* | 0.68 | 0.56 | 0.009 |
| 2 | -96 | -0.37 | 0.00 | -0.20 | -0.11 | 0.63* | 0.57 | 0.41 | 0.04 |
| 3 | -24 | -0.62 | 0.04 | 0.43* | -0.26* | 0.42 | 0.73 | 0.63 | 0.004 |
| 3 | -48 | -0.72* | 0.18 | 0.34* | -0.05 | 0.26 | 0.56 | 0.43 | 0.033 |