# Peer review of "Characterising spatio-temporal variability in seasonal snow cover at a regional scale from MODIS data: The Clutha Catchment, New Zealand"

_Hydrology and Earth System Sciences, 2019_

## Referee Comment (RC1) · Anonymous Referee #1 · 28 Feb 2019

General comments:

The authors present a detailed regional snow cover climatology and analysis of spatiotemporal variation for a subset of New Zealand. After creating a cloud-gap-filled daily time series of snow covered area, they extract several SCA derived metrics (snow cover duration, snow line elevation, percent basin at various SCA thresholds) to analyze spatial and temporal patterns and trends in snow. They then compare SCA derived metrics to several climatic metrics and indices (precipitation, temperature, winter air flow, SOI and SAM). The use of rPCA to explore spatial variability of SCD and the use of HYS-

[Figure]

PLIT to characterize wind trajectories is novel to the regional analysis of SCA patterns. Results show highly variable intra-annual and inter-annual spatiotemporal variability in snow covered area for this catchment with no significant trends in snow covered duration during the record analyzed. rPCA shows utility in identifying modes of spatial variability in snow cover duration with modes relating to both topographic and climatic influences. Temperature appears to be of greater influence at higher elevations in the study area, and precipitation of greater influence lower down, contrasting previous research (MoraÌĄn-Tejeda et al., 2013; Hammond et al., 2018). While climate oscillation indices show little skill in explaining snow variability, easterly winds appear to increase SCD in this region. Figures are of a high quality and effectively communicate major results of the study.

MoraÌĄn-Tejeda, E., LoÌĄpez-Moreno, J. I. and Beniston, M.: The changing roles of temperature and precipitation on snowpack variability in Switzerland as a function of altitude, Geophysical Research Letters, 40(10), 2131–2136, 2013.

Hammond J. C., Saavedra F. A., Kampf S. K.: Global snow zone maps and trends in snow persistence 2001–2016. Int J Climatol, 38, 4369–4383, https://doi.org/10.1002/joc.5674, 2018.

Specific comments:

A few outstanding questions and points remain following the author's summary of their methods and discussion:

Methods:

What is the rationale behind the 77% threshold used to show first 6 principal components?

A brief summary should accompany Jobst references explaining how the datasets were created (temperature: spline interpolation with a lapse rate model, precipitation...) so that the reader doesn't have to go to Jobst publications to figure this out.

How does SCA from MODImLab C6 and MOD10A1 C6 generally compare?

Why use parametric linear regression trend analysis with assumptions below instead of non-parametric Mann kendall analysis? -Linear relationship -Multivariate normality -No or little multicollinearity -No auto-correlation -Homoscedasticity

Why not compare wind trajectories to annual SCD, basinwide SCA %'s instead of comparing to principal components?

Can you conduct correlations between PC's and topographic, climatic metrics to have better understanding of what each PC represents?

Map of annual cloud persistence, longest cloud duration could aid in assessing uncertainty of Dozier et al. (2008) method for cloud-gap filling.

What is the data source for the SAM and SOI time series?

Discussion:

In explaining why easterly winds lead to greater snow in this region, are winds from the east more moisture laden? Associated with cooler temperatures?

Introduction mentions importance of greater understanding of snow distribution, and how it might change, to water resource management. How might the results of this work specifically be used to improve hydrologic modeling of snowmelt in this watershed and within sub-watersheds?

The authors attribute greater sensitivity of SCD to temperature at higher elevations to importance of rain snow threshold and importance of precipitation at lower elevations to precipitation that arrives during cold fronts. This rationale should be flushed out further with examples from other regions with similar divide-specific behavior, as the results contrast the results of previous work in Europe, North America, South America and Asia showing greater importance of precipitation at high elevations and in arid areas and temperature at low elevations (MoraĺĄn-Tejeda et al., 2013; Hammond et al.,
2018). How does P/PET map across the catchment? Likely relates to solar insolation?

Further discussion of why elevation-snow relationships degrade further away from the main divide would help strengthen this point.

Technical corrections, typing errors, etc.:

Snowmelt and snow melt used in paper, chose one and be consistent

Suggest avoiding "snow persistence" term to describe snow cover duration, as this phrase has become a metric of its own in the past decade

Sentences in abstract could be better interwoven to present results more effectively

End page 1 to start page 2 - Second sentence of introduction - Put references for first part of sentence corresponding to that point before comma, then references for second part of sentence at end. Similar comments on other sentences when all references displayed at end of multi-part sentence

Line 16 page 2. Suggest rephrasing sentence. Also display area normalized annual flow (mm/y) to help reader understand relative amount of flow

Line 31 page 3. First two sentences of this paragraph disjoint, could use topic sentence

Line 26 page 4. Reword to say MODIS useful for snow mapping. MODIS also "particularly" good for vegetation analyses etc...

Line 19 page 9. Mean temperature and total precipitation

Line 10 page 11. ", and ."

Line 5 page 12. "in the catchment At moderate"

Line 7 page 13. "With the inverse being true..." not needed since you have positive (negative)

Line 4 page 15. Sensitivity reduces, but not at a negative rate, suggest rephrasing Line

1 page 18. Remove "by"

Figure 11 - Could be moved to supplemental materials

Figure 14 - Consider another color than yellow for eastern mtn ranges. Yellow not very visible in electronic and printed versions

Figure 15 - What does this plot look like with elevation axis instead of distance to main divide? May help discussion on P vs T importance at different elevations

Figure 18 - Could be supplemental

Tables 1,3,5 - Could be supplemental

Table 2, Figures S9 and S32 from Hammond et al., 2018 may be apt comparisons to this study

---

## Referee Comment (RC2) · Markus Hrachowitz (Referee) · 20 May 2019

The manuscript "Characterising spatio-temporal variability in seasonal snow cover at a regional scale from MODIS data" by Redpath et al. provides a timely and detailed analysis of snow duration and snow covered area in a part of the world where quantitative knowledge on these variables is scarce. The manuscript will thus be of interest to many in the community. The paper is organized in a logical way, well-written and the analysis is based on sound methods. Overall I really enjoyed reading this paper and besides a few small notes, I do only have three major questions for and comments to

the authors:

(1) At the end of the introduction it remains somewhat vague what the actual objective of the paper is. The term "snow cover climatology" is very vague and can be interpreted in a wide range of ways. I am not even sure if the analysis provided in this manuscript show a robust enough link to "snow cover climatology". In addition, it remains unclear what the actual research question is and which specific research hypotheses are tested in this manuscript. To help the reader to better appreciate this work, I would thus strongly encourage the authors to (a) be more specific in their terminology and (b) explicitly formulate a set of research questions and the related research hypotheses.

(2) I am not sure if the use of the Coefficient of Variation (CV; Eq.1; section 4.3) is the most informative way to illustrate the spatial differences in annual snow cover fluctuations. As it is the ratio of the standard deviation over the mean, it will be strongly controlled by the mean. Thus, for example low elevations with short SCD will inherently show a bias towards higher CV than higher elevations. This is exactly what can be seen in figure 4b. I think the standard deviation would be much more informative to show here, because in the end it is all about the absolute change in snow covered days – in a region with only 5 snow covered days per year a standard deviation of say 2 days will result in a high CV – but in that region snow is rather irrelevant anyway. In contrast, in a region with 150 snow covered days the same standard deviation of 2 days will result in a much lower CV.

(3) It is not entirely clear why in section 3.3.3 suddenly the regional snowline elevation technique by Krajci et al. (2014) is used. This method is a spatial cloud filter. As I understand, here, clouds are filtered using a temporal filter (section 3.2.1). thus, for each pixel on each day, a binary value of snow yes/no is then available. In conjunction with a DEM, then also the elevation with snow in is known at that location – thus no need for further filtering. Please clarify this.

Minor points:

[Figure]

Section 2: please add a bit more information about the region, such as elevation range, mean annual and mean winter precipitation as well as temperatures or % glacier area

p.4, l.5: not sure what is meant by "water management is not a new concept..." – please rephrase.

p.4, l.32: as far as I know is the effective resolution of MODIS currently estimated at ∼2000m, due to factors such as topographic gradients. If I am not mistaken, the information content of the 500m product is already rather limited. Not sure if further downscaling is really warranted here. Please double-check!

p.5, l.6: please be more specific! How much higher are these resolutions?

p.5, l.10-11: please be specific! What does "rigorous" mean here? How do they do it?

p.5, l.23: what is meant by "data cube"?

p.5, l.24, figure 2: flow chart is nice, but please specify what the lighting symbols stand for. Potential problems?

p.5, l.32: what is meant by "erosion" here?

p.6, l.3-7: what are the largest gaps filled here? one day? One week? One month? Which length of gap would be considered unacceptable? Why?

p.7, l.3: not sure if "epoch" is a suitable term here. please rephrase.

p.8, l.15: on basis of what were the individual mountain ranges defined?

p.11, l.5: what do you exactly mean by "the typical snow cover climatology..."? please try to be more specific and rephrase that.

p.11, l.7, figure 4: the 120-days line is not really well-visible in the figure. Please also adjust the contrast in the color scheme in figure 4a – right now it looks as if essentially all the region has a SCD of ∼90 days.

p.11, l.10: should read as "...elevation of 615m. On average,..."

p.11, l.11: should this perennial snow cover than not be defined as a glacier?

p.11, l.13: ". . .appears" sounds very speculative here. why not directly linking it to and highlighting the position of these locations in figure 6?!

p.17, l.32-33: what is meant by "positive" and "negative" variance?

p.19, l.18-32: the reference Gao et al. (2017) would fit nicely here.

References: Gao, H., Ding, Y., Zhao, Q., Hrachowitz, M., & Savenije, H. H. (2017). The importance of aspect for modelling the hydrological response in a glacier catchment in Central Asia. Hydrological processes, 31(16), 2842-2859.
* * *

---

## Author Comment (AC3) · 18 Jun 2019

We thank the two referees for their thorough and helpful comments. Response to their feedback is presented below.

**1. Response to RC1 comments**

*General comments:*

*The authors present a detailed regional snow cover climatology and analysis of spatiotemporal variation for a subset of New Zealand. After creating a cloud-gap-filled daily time series of snow covered area, they extract several SCA derived metrics (snow cover duration, snow line elevation, percent basin at various SCA thresholds) to analyze spatial and temporal patterns and trends in snow. They then compare SCA derived metrics to several climatic metrics and indices (precipitation, temperature, winter air flow, SOI and SAM). The use of rPCA to explore spatial variability of SCD and the use of HYSPLIT to characterize wind trajectories is novel to the regional analysis of SCA patterns.*

*Results show highly variable intra-annual and inter-annual spatiotemporal variability in snow covered area for this catchment with no significant trends in snow covered duration during the record analyzed. rPCA shows utility in identifying modes of spatial variability in snow cover duration with modes relating to both topographic and climatic influences. Temperature appears to be of greater influence at higher elevations in the study area, and precipitation of greater influence lower down, contrasting previous research (MoraÌ€A¸n-Tejeda et al., 2013; Hammond et al., 2018). While climate oscillation indices show little skill in explaining snow variability, easterly winds appear to increase SCD in this region. Figures are of a high quality and effectively communicate major results of the study.*

*Specific comments:*

*A few outstanding questions and points remain following the author's summary of their methods and discussion:*

*Methods:*

*What is the rationale behind the 77% threshold used to show first 6 principal components?*

**Response**: The first six principal components were the focus of further analysis, as together they explained more than 75% of the observed variability. Note that the caption for Figure 11 contained a typo when describing the red line indicates 77% of explained variance, when it should have been 75. This has been corrected. The goal of PCA is to reduce the dimensionality of a dataset whilst identifying common modes of variability. In this case, the first six principal components, of a possible 16 (when all years are considered as factors), explain a substantial proportion of the variance, while excluding remaining components that individually contribute only a small, and increasingly noisy, component of the observed signal. PC6 is the first of the components that explains less than 5% of observed variability. The remaining components, being increasingly less spatially coherent are of limited use for interpretation. We have added detail regarding this at the end of Section 3.4.1

*A brief summary should accompany Jobst references explaining how the datasets were created (temperature: spline interpolation with a lapse rate model, precipitation...) so that the reader doesn't have to go to Jobst publications to figure this out.*

**Response:** Agreed, we have added detail regarding the interpolations method in the relevant section.

*How does SCA from MODImLab C6 and MOD10A1 C6 generally compare?*

**Response:** This is a relevant question but outside the scope of this paper. A comprehensive comparison can be found in Masson et al. (2018) with justification of the development of MODImLab for New Zealand applications by Sirguey et al. (2009). Nevertheless, we have made general comparisons between MODImLab and MOD10A1 initially as part of this work, and have found that while both datasets provide similar signals, MOD10A1 tends to over-estimate SCA within the Clutha Catchment. This is generally attributed to the presence of cloud, and misclassification of cloud, and/or cloud shadow, as snow by MOD10A1, a pitfall that has justified MODImLab in the first place. MOD10A1 also features aliasing artefacts due to reprojection from the SIN system in NZ all add considerable noise to sca maps, particularly around the snowline. The ability to map SCA at 250m with MODImLab further justified its use instead of MOD10A1.

*Why use parametric linear regression trend analysis with assumptions below instead of non-parametric Mann kendall analysis? -Linear relationship -Multivariate normality -No or little multicollinearity -No auto-correlation –Homoscedasticity*

**Response**: Our main motivation was to determine whether any obvious first order trends were present within the time series of snow cover metrics. In terms of assessing trends, non-parametric approaches such as the Theil-Sen slope approach, were considered alongside linear regression, and the results were consistent with those provided by linear regression. The lines of the linear fit and T-S fit have been added to Figure 1 to demonstrate this for the testing for a trend in winter median snow line. In this case, neither fit (applied to the annual medians) departs significantly from the series median. We appreciate that the Mann-Kendall test is popular to detect trend in time-series, however we must stress that parametric approaches are more powerful in general, and that the merit of the Mann-Kendall test is rather to be resilient to non-normally distributed series and outliers. Our trend analysis is done on the yearly median of the SLE. The sample distribution of the median is approximately normal and the median itself is already resilient to outliers. We therefore don't see much merit of using a less powerful non-parametric test such as Mann-Kendall in this case.

[Figure]

*Figure 1: Box plot of annual winter snow line elevation (SLE) with linear and Theil-Sen fits applied to the series of annual median values.*

*Why not compare wind trajectories to annual SCD, basinwide SCA %'s instead of comparing to principal components?*

**Response:** Once specific modes of spatial variability, associated with one or more years, were identified, our interest was in determining whether there were any strong climatic drivers associated with those spatial modes. We are interested in the impact of climate variability on spatial and temporal variability in SCA/SCD. Comparing basin wide metrics with wind trajectories would obscure the spatial component of variability. The advantage of PCA is that by considering the maps of spatial modes, and the loadings of those for each year, the bulk of spatio-temporal variability is preserved and considered together with climate variability.

*Can you conduct correlations between PC's and topographic, climatic metrics to have better understanding of what each PC represents?*

**Response**: We agree that an association between PCs and elevation could be expected. This was assessed early on in the analysis, but was found to not fully explain the spatial characteristics of PCs. As discussed in Section 5.3, PC1 is most strongly associated with elevation, mainly in terms of most of the variability in snow cover duration occurring for higher elevation areas where most of the snow cover exists. Elevation is also convolved with aspect, however, as indicated in Figure 7 in the manuscript. Furthermore, PC1 explains only 38% of the observed variability, indicating that additional factors beyond first order topographic controls play a substantial role. PC's 2 – 6 all demonstrate spatial patterns that are not related to topography in a straightforward way.

As suggested by the reviewer, the motivation of the final stage of our analysis, where correlations between PCs, and SOI, SAM and wind trajectory anomalies is to understand the relationship between PCs and larger scale atmospheric processes.

*Map of annual cloud persistence, longest cloud duration could aid in assessing uncertainty of Dozier et al. (2008) method for cloud-gap filling.*

**Response:** We agree with the reviewer's suggestion. We have added maps of mean cloud duration and maximum cloud duration, as well as a histogram of both metrics. While cloud covered periods are frequent at the pixel level, they are also relatively short in general. The area with the most persistent cloud cover is on and near the main divide, where snow is also most persistent, and the south eastern margin of the catchment, which is below the seasonal snow zone. Ultimately, the areas where snow is most variable, and more frequent observations are necessary, are less affected by cloud cover.

*What is the data source for the SAM and SOI time series?*

**Response:** Data for SAM and SOI were sourced from NOAA CPC, The URLs corresponding to those data are now provided in the revised manuscript.

*Discussion:*

*In explaining why easterly winds lead to greater snow in this region, are winds from the east more moisture laden? Associated with cooler temperatures?*

**Response:** The rationale here is that synoptic circulation affecting this part of NZ is generally dominated by strong westerly flow, which is intercepted by the main divide of the Southern Alps, at the western boundary of the catchment. This results in extremely high precipitation in the west of the catchment, but a strong eastward precipitation gradient to the east, and an associated rain shadow effect in the east. When easterly flow occurs, regions of the catchment that are typically

sited within the rain shadow become the first to intercept moist airmasses originating from the east. The storms that may bring such conditions to the South Island can be cold, but this is not always the case. Under these conditions, western parts of the catchment become drier (than normal). We have reworked the text in Section 5.4.3 to clarify this.

*Introduction mentions importance of greater understanding of snow distribution, and how it might change, to water resource management. How might the results of this work specifically be used to improve hydrologic modeling of snowmelt in this watershed and within sub-watersheds?*

**Response**: Snow modelling efforts in New Zealand have historically been underpinned by degree-day models, and more recently enhanced degree-day models. These models often underperform at large scales, and their refinement is the subject of ongoing research. This work provides insight in the spatial variability in sensitivity of SCD to variability in temperature and precipitation, which can in turn inform the variable performance of such models. This work provides insight into the need to better capture the processes controlling seasonal snow in order to improve models, and in the local context, provides guidance on where gains can be made. Furthermore, the snow climatology, and time series of snow products produced through this work provide a useful tool for the validation of snow models. We have added additional detail regarding these points in Section 5.4.2.

*The authors attribute greater sensitivity of SCD to temperature at higher elevations to importance of rain snow threshold and importance of precipitation at lower elevations to precipitation that arrives during cold fronts. This rationale should be flushed out further with examples from other regions with similar divide-specific behavior, as the results contrast the results of previous work in Europe, North America, South America and Asia showing greater importance of precipitation at high elevations and in arid areas and temperature at low elevations (MoraÌÀ¸n-Tejeda et al., 2013; Hammond et al., 2018).*

**Response:** It appears that the reviewer may have misinterpreted our results here. Our intention was not to frame variability in sensitivity of SCD to either temperature or precipitation in terms of elevation, but rather to point out the high degree of spatial variability across the catchment. The updated Fig. 15 will hopefully emphasise this, where the improved explanatory power of distance from the main divide relative to elevation is obvious. Indeed, as figure 15 shows, sensitivity to T is greatest at lower elevations, but this relationship is relatively weak compared to distance to the main divide. We have reworked Section 4.4.1 of the results to clarify this, with respect to Fig. 15. We have also emphasised this point in Section 5.4.3. Indeed, the results presented here are broadly consistent with those of Hammond et al., (2018), but allow us to further deconvolve the meridional gradients in sensitivity across the catchment, which are not as readily apparent when space is considered only in terms of latitude and elevation. We have also further stressed these points, and contextualised them with respect to Hammond et al., 2018, in Section 5.4.2.

*How does P/PET map across the catchment? Likely relates to solar insolation?*

As described in Section 2, there is a strong precipitation gradient across the catchment, from west to east. It is pointed out in Section 5.4.2, with reference to Macara (2015), that solar irradiance is highest in the central part of the catchment, where sensitivity to temperature and precipitation is reduced. The role of solar radiation is subject to ongoing work locally, especially given the variable performance of P and T as predictors of SCD, as revealed here.

*Further discussion of why elevation-snow relationships degrade further away from the main divide would help strengthen this point.*

**Response:** Currently, there is a lack of data to form, analyse, and support a range of hypotheses that could explain the degradation in the snow-elevation relationship away from the main divide. Our work here identified and characterised this behaviour. In the manuscript we discuss one hypothesis; that a change in morphology of mountain ranges, from the alpine in nature Southern Alps, to the Fault-Block ranges of Central Otago may increase the efficacy of the wind in redistributing snow from relatively broad, high and flat accumulation areas into steep sided incised gullies and other terrain features. We believe however that more work is needed beyond the scope of this paper, to explore fully this and potentially raise other hypotheses.

*Technical corrections, typing errors, etc.:*

*Snowmelt and snow melt used in paper, chose one and be consistent*

**Response**: Agreed, uses of snow melt replaced with snowmelt.

*Suggest avoiding "snow persistence" term to describe snow cover duration, as this phrase has become a metric of its own in the past decade*

**Response:** Agreed, snow persistence replaced with SCD where appropriate.

*Sentences in abstract could be better interwoven to present results more effectively*

**Response:** Agreed, the abstract has been reworked.

*End page 1 to start page 2 - Second sentence of introduction - Put references for first part of sentence corresponding to that point before comma, then references for second part of sentence at end. Similar comments on other sentences when all references displayed at end of multi-part sentence*

**Response:** Agreed, the sentence has been fixed according to the reviewer's suggestion.

*Line 16 page 2. Suggest rephrasing sentence. Also display area normalized annual flow (mm/y) to help reader understand relative amount of flow*

**Response:** Agreed, the area normalised flow has been added, and the sentence rephrased.

*Line 31 page 3. First two sentences of this paragraph disjoint, could use topic sentence*

**Response:** Agreed, the sentences were reworked.

*Line 26 page 4. Reword to say MODIS useful for snow mapping. MODIS also "particularly" good for vegetation analyses etc...*

**Response**: Agreed, the sentence has been modified as per the reviewer's comment.

*Line 19 page 9. Mean temperature and total precipitation*

**Response**: Agreed, the sentence has been sentence modified.

*Line 10 page 11. ", and ."*

**Response:** Removed *", and"*.

*Line 5 page 12. "in the catchment At moderate"*

**Response:** We added the missing period at the end of the sentence.

*Line 7 page 13. "With the inverse being true..." not needed since you have positive (negative)*

**Response:** removed *"With the inverse being true..."*

*Line 4 page 15. Sensitivity reduces, but not at a negative rate, suggest rephrasing*

**Response:** Agreed, the negative sign has been removed from the value of 0.43 to avoid the double negative.

*Line 1 page 18. Remove "by"*

**Response:** This is not a "by", but a "hy" meaning hydrological year, as defined in Section 3.2.1.

*Figure 11 - Could be moved to supplemental materials*

**Response:** We agree, and have moved Fig. 11 to the supplement.

*Figure 14 - Consider another color than yellow for eastern mtn ranges. Yellow not very*

*visible in electronic and printed versions*

**Response**: The colours of the bars in Figure 13 and 14 have been updated.

*Figure 15 - What does this plot look like with elevation axis instead of distance to main*

*divide? May help discussion on P vs T importance at different elevations*

**Response:** We agree, this modification can help illustrate the role of P vs T. Figure 15 (now 16) has been amended to include subplots of the sensitivity of SCD anomaly to variability in T and P as a function of both elevation and distance to the main divide. It is now clear for the reader that while there is a (relatively weak) relationship between elevation and sensitivity to temperature, the relationship is much stronger for distance from the main divide. In terms of precipitation, there is no relationship between elevation and precipitation sensitivity, while there is a (ultimately non-linear) relationship between precipitation and distance to the main divide.

*Figure 18 - Could be supplemental*

**Response:** We have retained Figure 18 as it relates to key discussion points concerning persistent winter anticyclonic conditions affecting southern New Zealand, as well as demonstrating the general warming trend observed at AWS across the catchment, which is consistent with observations in New Zealand more generally.

*Tables 1,3,5 - Could be supplemental*

**Response:** We agree regarding Tables 1 and 5, and have moved these to the Supplement. We have retained Table 3 as we find it useful for assessing the groupings of years within the time series.

*Table 2, Figures S9 and S32 from Hammond et al., 2018 may be apt comparisons to this study*

**Response:** Agree, we have included comparisons to Hammond et al., (2018) in Sections 5.4.2 and 5.4.3..

**2. Response to RC2 comments**

*The manuscript "Characterising spatio-temporal variability in seasonal snow cover at a regional scale from MODIS data" by Redpath et al. provides a timely and detailed analysis of snow duration and snow covered area in a part of the world where quantitative knowledge on these variables is scarce. The manuscript will thus be of interest to many in the community. The paper is organized in a logical way, well-written and the analysis is based on sound methods. Overall I really enjoyed reading this*

*paper and besides a few small notes, I do only have three major questions for and comments to the authors:*

*(1) At the end of the introduction it remains somewhat vague what the actual objective of the paper is. The term "snow cover climatology" is very vague and can be interpreted in a wide range of ways. I am not even sure if the analysis provided in this manuscript show a robust enough link to "snow cover climatology". In addition, it remains unclear what the actual research question is and which specific research hypotheses are tested in this manuscript. To help the reader to better appreciate this work, I would thus strongly encourage the authors to (a) be more specific in their terminology and (b) explicitly formulate a set of research questions and the related research hypotheses.*

**Response:** We thank the reviewer for this constructive comment. The final paragraphs of the introduction have been reworked to more clearly state the research aims, and to provide an explicit definition of "snow cover climatology" in this context.

*(2) I am not sure if the use of the Coefficient of Variation (CV; Eq.1; section 4.3) is the most informative way to illustrate the spatial differences in annual snow cover fluctuations. As it is the ratio of the standard deviation over the mean, it will be strongly controlled by the mean. Thus, for example low elevations with short SCD will inherently show a bias towards higher CV than higher elevations. This is exactly what can be seen in figure 4b. I think the standard deviation would be much more informative to show here, because in the end it is all about the absolute change in snow covered days – in a region with only 5 snow covered days per year a standard deviation of say 2 days will result in a high CV – but in that region snow is rather irrelevant anyway. In contrast, in a region with 150 snow covered days the same standard deviation of 2 days will result in a much lower CV.*

**Response:** While the standard deviation would also provide insight into variability, The CV is used here as it provides a means to visualise pixel-wise dispersion about the mean SCD across the catchment, which was our main motivation. It also acts to highlight areas where low frequency, high magnitude events (e.g., disruptive, low level and long-lasting snow events) can occur, while snow is not always relevant in these areas, but can have a significant impact when it occurs. The standard deviation suggests spatial anomalies near the 120-day snow contour in particular, where high spatial variability in absolute terms is seen to be more consistent with adjacent areas when expressed as CV. The CV still retains sufficient signal to highlight the relative variability in snow seasons both as a function of elevation and aspect (Figure 7 in the manuscript). We acknowledge that the CV map is most useful when considered at the same time as corresponding map of the mean. We also note that CV is to characterised variability in other similar studies (e.g., Hammond et al., 2018). We have added justification and references for the use of CV in the text in Section 3.3.

[Figure]

*Figure 2: Maps of the CV (a) and standard deviation (b) of snow cover duration for the Clutha Catchment.*

*(3) It is not entirely clear why in section 3.3.3 suddenly the regional snowline elevation technique by Krajci et al. (2014) is used. This method is a spatial cloud filter. As I understand, here, clouds are filtered using a temporal filter (section 3.2.1). thus, for each pixel on each day, a binary value of snow yes/no is then available. In conjunction with a DEM, then also the elevation with snow in is known at that location – thus no need for further filtering. Please clarify this.*

**Response:** While it is true that Kracji et al., (2014) propose their snowline method as a potential approach to cloud filtering (as long as cloud cover is not excessive), it also offers a more fundamental use in robustly determining a regional snow line elevation in a relatively efficient way. While the snowline could be inferred from basin-wide snow covered area and the basin hypsometry, this approach can be inaccurate when the snowline is highly variable (as a function of aspect, for example, commonly a difference in snowline elevation of 1000 m or more can be observed between north and south aspects in this region), or when localised snow fall lowers the SLE substantially, but locally, at a sub-basin scale. In this case, Krajci et al.'s method is substantially more robust and we find it relevant to determine SLE even if issues with cloud have been addressed in another step. In the case of the median winter snowline of 1270 m, for example, that was found to correspond with basin-wide SCA ranging from 3800 km2 to more than 5000 km2. Simply inferring SLE from SCA and hypsometry would, in these cases, potentially lead to large over and under estimations of SLE under varying conditions.

*Minor points: Section 2: please add a bit more information about the region, such as elevation range, mean annual and mean winter precipitation as well as temperatures or % glacier area*

**Response:** Agreed, we have added details as per the reviewer's suggestion.

*p.4, l.5: not sure what is meant by "water management is not a new concept. . ." – please rephrase.*

**Response:** This sentence has been reworded.

*p.4, l.32: as far as I know is the effective resolution of MODIS currently estimated at ~2000m, due to factors such as topographic gradients. If I am not mistaken, the information content of the 500m product is already rather limited. Not sure if further downscaling is really warranted here. Please double-check!*

**Response**: The resolution of 250 m is achieved following the fusion of the specific MODIS bands used by MODImLab for spectral unmixing, with merit, benefits, and validation discussed in detailed in Sirguey et al. (2008, 2009).

*p.5, l.6: please be more specific! How much higher are these resolutions?*

**Response:** Agreed, we have specified the higher (15m) spatial resolution of the ASTER sensor.

*p.5, l.10-11: please be specific! What does "rigorous" mean here? How do they do it?*

**Response:** We have added more detail here on the topographic correction in particular, which is central to the successful mapping of snow in rugged alpine terrain.

*p.5, l.23: what is meant by "data cube"?*

**Response:** Agreed, we have added a definition of the term *"data cube"*.

*p.5, l.24, figure 2: flow chart is nice, but please specify what the lighting symbols stand for. Potential problems?*

**Response:** The lightning symbols represent processing steps – this has been clarified in the figure.

*p.5, l.32: what is meant by "erosion" here?*

**Response:** In this case we are referring to morphological erosion, a relatively common image processing technique which is useful for removing spurious (in this case, mis-classified) pixels around the edge of a feature, while still retaining the features shape. We have clarified in the text that it is morphological erosion being used and have added a reference.

*p.6, l.3-7: what are the largest gaps filled here? one day? One week? One month? Which length of gap would be considered unacceptable? Why?*

**Response:** We agree with the reviewer this is a relevant question. It was also pointed out by the first reviewer. We have added maps and a histogram of the mean and maximum duration of cloud gaps, and provided detail in the text. Averaged over the whole time series, the mean consecutive cloud duration for each pixel is 2.7 days. Considering the maximum duration of cloud cover for pixel for each year, the average over the time series is 12.1 days. Obviously extended periods with no snow cover information from the MODIS imagery will weaken the interpolation, yet an average of a clear view at least once every five days will ensure that the interpolation is well constrained. In terms of the maximum cloud duration, extremes (>20 days) are limited to small areas on the main divide. These are also the areas where variability in snow cover is least, somewhat mitigating the impact of

less frequent clear views. Areas with maximum cloud duration > 15 days are also fairly spatially limited.

*p.7, l.3: not sure if "epoch" is a suitable term here. please rephrase.*

**Response:** Agreed, we replaced this term by "a point in time".

*p.8, l.15: on basis of what were the individual mountain ranges defined?*

**Response:** The mountain ranges were defined primarily according to the naming conventions from the authoritative NZ Geographic Names topographic dataset, form the national topographic database. A DEM and stream centreline network were used to provide guidance for the manual delineation between adjacent ranges. While this could be considered a relatively subjective approach, it provides a set of geographic units that are familiar to stakeholders within the basin.

*p.11, l.5: what do you exactly mean by "the typical snow cover climatology. . ."? please try to be more specific and rephrase that.*

**Response:** "Typical" is now deleted, as its use was redundant. We have added a clearer definition of "snow cover climatology" into the introduction. While the 16 year record is short, the intent here is to characterise the snow cover climatology spatially, with the map of mean SCD (Figure 5(a)), and temporally, with the curve of of snow covered area and snow line elevation over the course of a hydrological year. This can be characterised by the mean daily snow covered area curve (Figure 6).

*p.11, l.7, figure 4: the 120-days line is not really well-visible in the figure. Please also adjust the contrast in the color scheme in figure 4a – right now it looks as if essentially all the region has a SCD of ~90 days.*

**Response:** The symbology of Fig. 4 has been adjusted.

*p.11, l.10: should read as ". . .elevation of 615m. On average,. . ."*

**Response:** This has been fixed.

*p.11, l.11: should this perennial snow cover than not be defined as a glacier?*

**Response**: Not necessarily. We have added a comment on how this area compares to recent mapping of glaciers within the catchment.

*p.11, l.13: ". . .appears" sounds very speculative here. why not directly linking it to and highlighting the position of these locations in figure 6?!*

**Response:** Agree, this has been reworded.

*p.17, l.32-33: what is meant by "positive" and "negative" variance?*

**Response**: agree that variance is not the appropriate term here and have changed to deviation.

*p.19, l.18-32: the reference Gao et al. (2017) would fit nicely here*

**Response:** We agree and have incorporated into Section 5.4.2 of the discussion.

---

## Author Response (AR2)

Dear Markus Hrachowitz

Thank you for your consideration and acceptance of our manuscript. Note we have amended the acknowledgments to include reviewers.

Regards,

Todd Redpath, on behalf of the authors